# Private–public mappings in human prefrontal cortex

**Dan Bang[1,2]\*, Sara Ershadmanesh[3], Hamed Nili[4], Stephen M Fleming[1,5,6]**

[1]Wellcome Centre for Human Neuroimaging, UCL, London, United Kingdom; [2]Department of Experimental Psychology, University of Oxford, Oxford, United Kingdom; [3]School of Cognitive Sciences, Institute for Research in Fundamental Sciences, Tehran, Islamic Republic of Iran; [4]Wellcome Centre for Integrative Neuroimaging, Centre for Functional Magnetic Resonance Imaging of the Brain, University of Oxford, Oxford, United Kingdom; [5]Max Planck UCL Centre for Computational Psychiatry and Ageing Research, UCL, London, United Kingdom; [6]Department of Experimental Psychology, UCL, London, United Kingdom

**Abstract** A core feature of human cognition is an ability to separate private states of mind – what we think or believe – from public actions – what we say or do. This ability is central to successful social interaction – with different social contexts often requiring different mappings between private states and public actions in order to minimise conflict and facilitate communication. Here we investigated how the human brain supports private-public mappings, using an interactive task which required subjects to adapt how they communicated their confidence about a perceptual decision to the social context. Univariate and multivariate analysis of fMRI data revealed that a private-public distinction is reflected in a medial-lateral division of prefrontal cortex – with lateral frontal pole (FPl) supporting the context-dependent mapping from a private sense of confidence to a public report. The concept of private-public mappings provides a promising framework for understanding flexible social behaviour.

**\*For correspondence:**
danbang.db@gmail.com

**Competing interests:** The authors declare that no competing interests exist.

## Introduction

A striking aspect of human social behaviour is how often we say things we do not really mean simply because the situation requires it. For example, a politician may have doubts about a policy but, in order to boost voters' faith in her, defend it with great confidence. Conversely, an employee may be certain that his manager is wrong but, in order to minimise social friction, tentatively present his argument. Central to these scenarios is a distinction between private states of mind – what we think or believe – and public actions – what we say or do. A general ability to separate private and public aspects of mental states is thought to have evolved under the constraints of social communication (*Dennett, 2017*). In having a 'control buffer' between private states and public actions, an agent may avoid revealing its current state to competitors (e.g., by suppressing signs of fragility). It also allows an agent to deceive competitors (e.g., by displaying signs of strength) without interfering with processes that are needed for resolving its current state (e.g., taking steps to replenish energy).

In modern life, where people navigate complex social worlds, this ability to adapt private-public mappings to the context takes on new functions – including minimising conflict and facilitating communication. For example, social norms pertaining to politeness vary between contexts, such that a response which is appropriate in one context may be very inappropriate in another (e.g., commenting on a person's appearance at home versus at work). Contextual modulation of private-public mappings is, however, particularly challenging: social norms may be arbitrary, social contexts are typically diffuse over time and space, and there is often a tension between the nature of our private

states and the public actions required by the context. Indeed, context-inappropriate social behaviour is a common symptom in a variety of neuropsychiatric conditions – such as frontotemporal dementia (*Ibañez and Manes, 2012*), autism (*Happé, 1994*), schizophrenia (*Penn et al., 2002*) and borderline personality disorder (*King-Casas et al., 2008*) – at times with profound effects on quality of life.

Understanding why and how social function is impaired in these conditions is likely to be aided by a more thorough characterisation of the cognitive and neural mechanisms that underpin contextual modulation of private-public mappings. Here we studied how the brain solves this problem using decision confidence as a model system. Confidence is well-suited for studying private-public mappings because there is often a dissociation between the confidence that we privately feel and that which we publicly communicate (*Aitchison et al., 2015*; *Bang et al., 2017*; *Hertz et al., 2017*). Further, recent experimental work has provided us with the tools to separately manipulate private and public aspects of confidence (*Bang et al., 2017*; *Bang and Fleming, 2018*). In our experiment, subjects were required to communicate their confidence about simple perceptual decisions in different social contexts, with each context requiring a different mapping from private to public confidence. For example, one context required subjects to overstate their confidence in order to maximise reward, whereas another required them to understate their confidence.

Preliminary evidence suggests that a private-public distinction for confidence is reflected in a medial-lateral division of prefrontal cortex (PFC). First, recent work has indicated that dorsal anterior cingulate cortex (dACC) (*Fleming et al., 2012*; *Fleming et al., 2018*) and perigenual anterior cingulate cortex (pgACC) (*Bang and Fleming, 2018*; *De Martino et al., 2013*; *Gherman and Philiastides, 2018*; *Lebreton et al., 2015*; *Wittmann et al., 2016*) – both located in the medial wall of PFC – support the formation of an internal (private) sense of confidence. Second, a large body of literature has observed activation of lateral PFC, in particular the lateral frontal pole (FPl), in relation to explicit (public) reports of confidence (*Bang and Fleming, 2018*; *De Martino et al., 2013*; *Fleming et al., 2012*; *Fleming et al., 2018*; *Gherman and Philiastides, 2018*; *Hilgenstock et al., 2014*; *Shekhar and Rahnev, 2018*; *Yokoyama et al., 2010*) – raising the possibility that lateral PFC supports a mapping between private and public confidence. This hypothesis fits with a broader role of lateral PFC in cognitive control functions such as task or set switching (*Badre, 2008*; *Badre and Nee, 2018*). It also fits with the observation that injury to lateral PFC is associated with context-inappropriate social behaviour (*Ibañez and Manes, 2012*). Here, by combining univariate and multivariate analyses of fMRI data, we provide evidence for this division of labour, showing that FPl supports the context-dependent mapping from an internal sense of confidence to a public report.

## Results

### Experimental manipulation of private-public mappings

Subjects (*n* = 28) performed a social perceptual decision task, first in a behavioural session and subsequently during fMRI (*Figure 1*). On each trial, subjects made a group decision about a visual stimulus with one of four partners. Subjects were told that the partners were created by replaying the responses of four people performing the perceptual task on a separate day but, in reality, the partners were simulated. First, subjects judged whether a field of dots was moving left or right. Next, after being informed about the identity of their partner on the current trial, subjects were asked to report their confidence in the perceptual judgement – an estimate which would enter into the group decision. Subjects were then shown the current partner's response for that trial. Finally, implementing a common group decision rule (*Bang et al., 2017*), the individual decision made with higher confidence was automatically selected as the group decision, after which feedback about its accuracy was delivered. Subjects were incentivised to help each group achieve as many correct group decisions as possible but, by design, could only affect group decisions through their confidence reports.

We varied two features of the task in a factorial (4 × 4) manner. First, we varied the fraction of coherently moving dots (coherence) to manipulate subjects' internal sense of confidence in a perceptual judgement. In general, the higher the coherence, the higher subjects' confidence. Second, we engineered the partners (context) to have the same choice accuracy as subjects but to differ in mean confidence. In this case, the strategy for maximising group accuracy (and thereby reward) is to match a partner's mean confidence (*Figure 1—figure supplement 1*). If a subject always reports higher confidence than a partner, then they would not benefit from the trials where they were wrong but

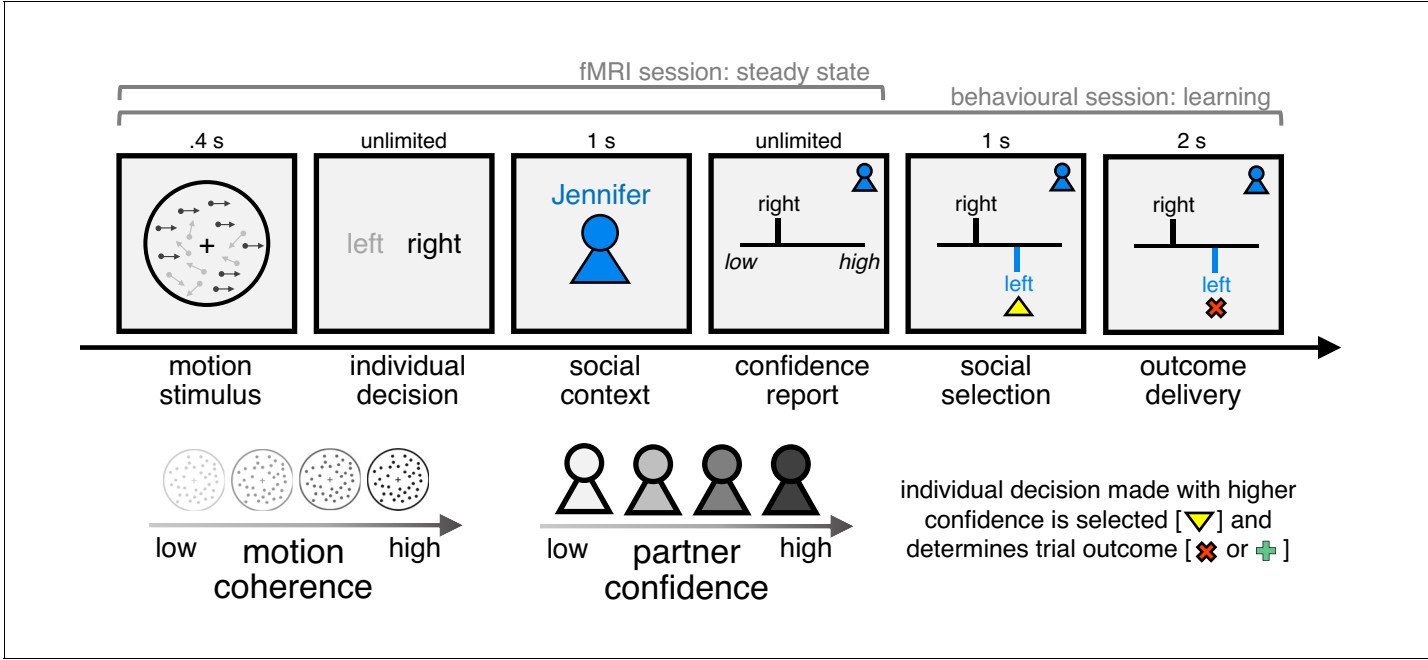

**Figure 1.** Experimental framework for dissociating private and public confidence. On each trial, subjects made a perceptual group decision with one of four partners. They first decided whether a random dot motion stimulus was moving left or right. We varied the fraction of coherently moving dots in order to manipulate subjects' internal sense of confidence in their decision. Subjects were then informed about their partner on the current trial and were asked to submit a report of confidence in their initial decision (discrete scale from 1 to 6). Subjects were then shown the partner's response, after which the individual decision made with higher confidence was selected as the group decision. Finally, subjects received feedback about choice accuracy, before continuing to the next trial. We engineered the partners to have the same choice accuracy as subjects but to differ in mean confidence. Subjects were incentivised to help each group achieve as many correct decisions as possible: they were told that we would randomly select two trials for each group in each session (4 × 2 × 2 = 16 trials) and pay £1 in bonus for every correct group decision (in reality, all subjects received £10 in bonus). In this design, the strategy for maximising group accuracy (reward) is to match your partner's mean confidence. The structure of the task differed between the behavioural and fMRI sessions as explained in the main text.

The online version of this article includes the following figure supplement(s) for figure 1:

**Figure supplement 1.** Confidence matching maximises group accuracy and thereby reward.
**Figure supplement 2.** Schematic of study protocol.
**Figure supplement 3.** Confidence distributions used to generate four partners who differ in mean confidence.

the partner was correct. In contrast, if a subject always reports lower confidence than a partner, then they would not benefit from the trials where the partner was wrong but they were correct. The design thus allowed us to separate public from private confidence and thereby probe private-public mappings in behaviour and brain activity.

The aim of the behavioural session was to calibrate levels of coherence so as to achieve target levels of choice accuracy and to train subjects on the social task (*Figure 1—figure supplement 2*). The training had three phases. First, subjects were paired with the four partners in a block-wise manner. Second, the partners were interleaved, with the identity of the current partner revealed after the perceptual choice. Finally, using the same design as in the fMRI session, the 'showdown' stage was played out in the background, with subjects not seeing the partner's response or the group outcome. This change, which was introduced to minimise inter-trial dependencies in behavioural and neural responses, meant that subjects had to rely on their knowledge (expectations) about the partners learned in the preceding behavioural session. In addition, to establish a baseline for behavioural and neural responses, we added a condition where the partner's identity was hidden. The fMRI session consisted of four scan runs, with the distribution of conditions matched across runs in order to facilitate multivariate analysis of the fMRI data (i.e. four trials for each coherence x context condition in each scan run). To help subjects keep track of the behaviour of each group, they were informed every 20 trials how often their individual decision had been selected as the group decision for a particular partner. These selection statistics were reset after each scan run.

## Confidence reports reflect motion coherence and social context

We first tested whether subjects' confidence reports in the fMRI session varied as a function of our factorial design (see *Figure 2—figure supplement 1* for analysis of prescan session). As intended, subjects' confidence reports were influenced by both motion coherence and social context (*Figure 2*, ordinal regression; coherence: $t(27) = 6.95$, p<0.001, context: $t(27) = 4.82$, p<0.001, interaction: $t(27) = -0.03$, p=0.975). More specifically, subjects' reported confidence increased with the level of coherence (*Figure 2A*) and the mean confidence of the current partner (*Figure 2B*). In other words, the confidence reported for a specific level of coherence depended on the current partner's mean confidence (*Figure 2C*) – demonstrating that subjects flexibly adjusted the mapping from an internal sense of confidence to an explicit report of confidence according to the social context.

## Encoding of motion coherence and social context in prefrontal cortex

We focused our fMRI analysis on three regions of interest (ROIs) that have been identified as putative neural substrates of decision confidence across a variety of studies but whose role in the generation of a context-dependent confidence report is unclear (*Figure 3A*). First, dACC and pgACC, located in the medial wall of PFC, have been consistently linked to the formation of an internal (private) sense of confidence (*Bang and Fleming, 2018*; *De Martino et al., 2013*; *Fleming et al., 2012*; *Fleming et al., 2018*; *Gherman and Philiastides, 2018*; *Lebreton et al., 2015*; *Wittmann et al., 2016*). For example, a recent fMRI study found that pgACC tracked all variables necessary for the formation of an internal sense of confidence in a novel psychophysical task that isolates decision confidence from its component parts (*Bang and Fleming, 2018*). Second, FPl, a region in human prefrontal cortex with no homologue in the monkey brain (*Neubert et al., 2014*), has consistently been found to track explicit (public) reports of confidence (*Bang and Fleming, 2018*; *Fleming et al., 2012*; *Fleming et al., 2018*; *Gherman and Philiastides, 2018*; *Hilgenstock et al., 2014*; *Shekhar and Rahnev, 2018*; *Yokoyama et al., 2010*). Of the three ROIs, FPl is of particular interest. We have previously hypothesised that FPl supports the mapping from private to public confidence (*Bang and Fleming, 2018*). First, FPl is not activated by tasks that vary private confidence in the absence of explicit reports (*Bang and Fleming, 2018*). Second, the microstructure (*Allen et al., 2017*; *Fleming et al., 2010*) and integrity (*Fleming et al., 2014*) of FPl predicts the degree to which an individual's confidence reports reflect their task performance – a relationship which may be

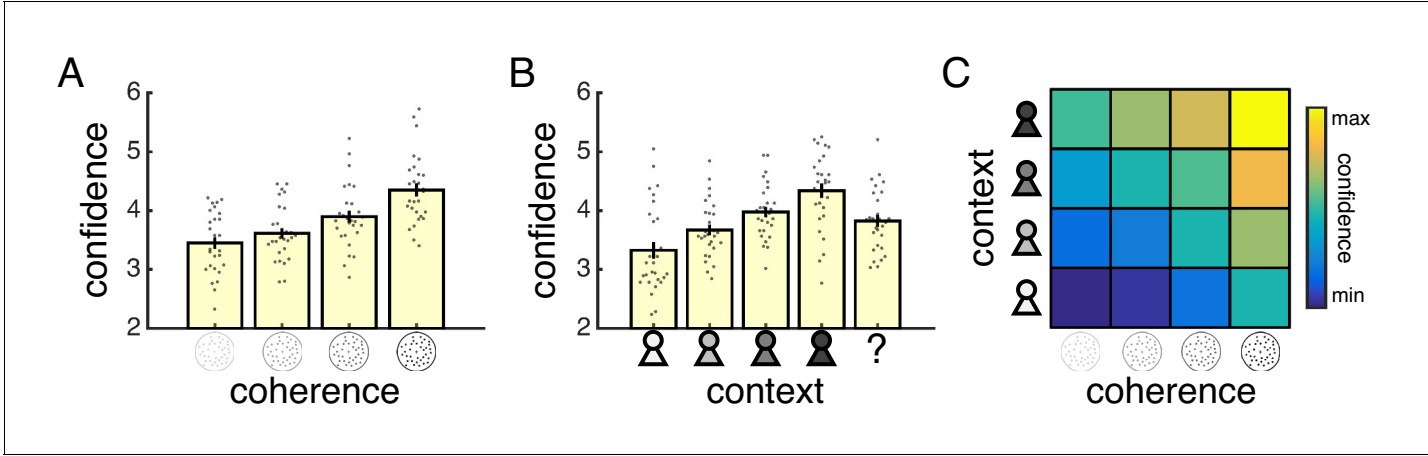

**Figure 2.** Separate influences of motion coherence and social context on confidence reports. (**A**) Mean confidence reported for each level of coherence. (**B**) Mean confidence reported when playing with each partner. The question mark indicates a condition where the partner's identity was hidden. (**C**) Heat map visualising mean confidence in each condition of our factorial design (confidence was z-scored for each subject before averaging across subjects). Warmer colours indicate higher confidence. All data are from the fMRI session. In (**A**) and (**B**), data are represented as group mean ± SEM. Each dot is a subject.

The online version of this article includes the following figure supplement(s) for figure 2:

**Figure supplement 1.** Analysis of confidence reports in prescan session.
**Figure supplement 2.** Analysis of confidence reports separated by partner identity.

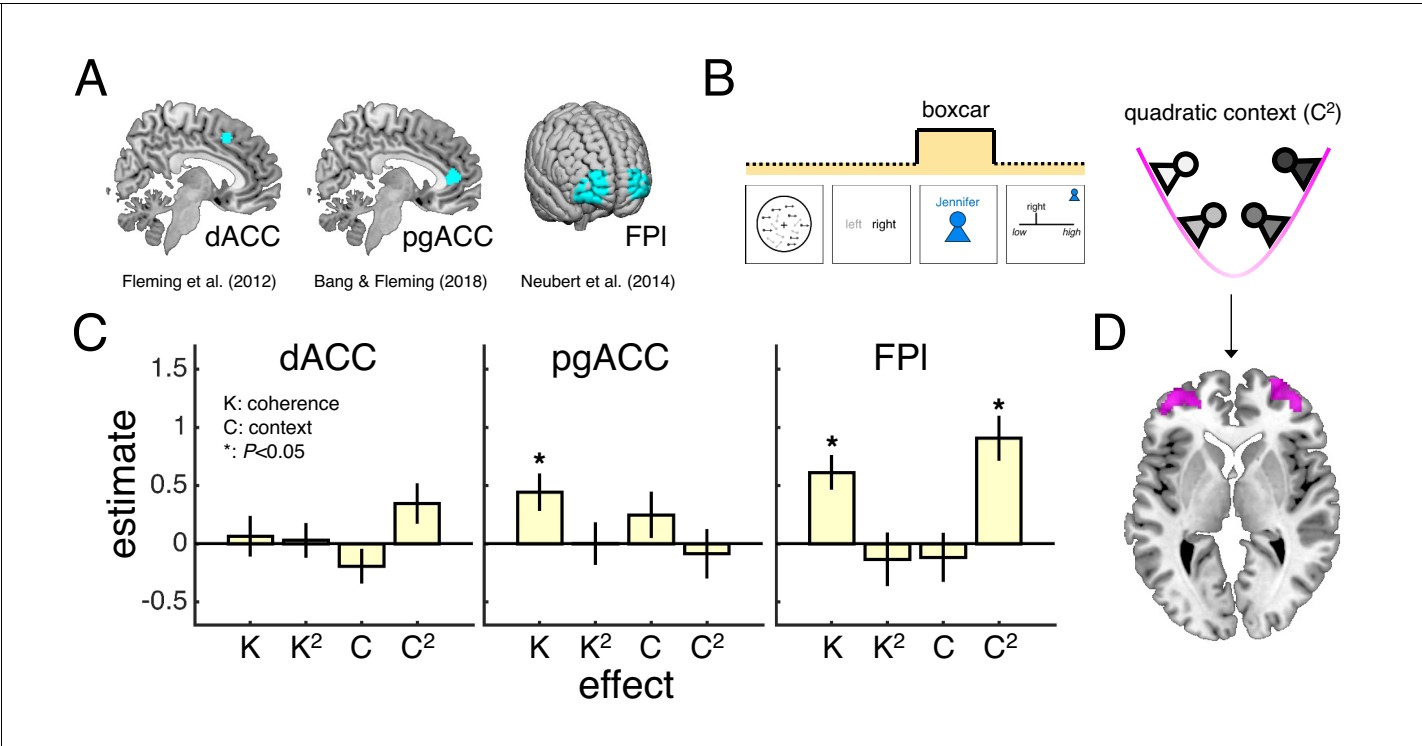

**Figure 3.** Encoding of motion coherence and social context in lateral frontal pole. (**A**) Regions of interest (ROIs). (**B**) We modelled neural responses to the context screen, including both linear and quadratic terms for coherence and context as parametric modulators – with the quadratic context term indexing the need for a context-dependent private-public mapping. (**C**) ROI contrast estimates for coherence (K), quadratic coherence (K$^2$), context (C) and quadratic context (C$^2$). We tested significance (asterisk) by comparing contrast estimates across subjects to zero (p<0.05, one-sample *t*-test). Statistical results are summarised in *Table 1*. Data are represented as group mean ± SEM. (**D**) Visualisation of whole-brain activation for quadratic context in lateral prefrontal cortex (clusters significant at p<0.05, FWE-corrected for multiple comparisons, with a cluster-defining threshold of p<0.001, uncorrected). See Appendix 1 for whole-brain activations in response to context screen and Appendix 2 for whole-brain activations in response to presentation of the motion stimulus. dACC: dorsal anterior cingulate cortex. pgACC: perigenual anterior cingulate cortex. FPl: lateral frontal pole. The online version of this article includes the following figure supplement(s) for figure 3:

**Figure supplement 1.** Evaluation of quadratic terms.

explained by a role of FPl in maintaining a stable mapping from private to public confidence. However, to date, this hypothesis about the function of FPl has never been directly tested.

As an initial assessment of the contribution of the three ROIs to a context-dependent mapping from private to public confidence, we first estimated a general linear model (GLM1) of neural activity locked to the screen that revealed the current partner (context screen; *Figure 3B*). We selected the context screen as our time window of interest for two reasons. First, the context screen is the first point in a trial that information about the current partner is revealed – any context-related regressors would have little meaning if assigned to timepoints earlier on in a trial. Second, during the presentation of the context screen, subjects have all the information needed to internally decide on a context-dependent confidence report, but the neural response will not be confounded by the engagement of motor processes needed to select a confidence report – a motor plan can only be prepared after the randomised initial location of the confidence marker is revealed. Building on our factorial design, we included linear and quadratic terms for coherence and context as parametric modulators of the neural response to the context screen. In this model, the quadratic context term captures the intuition that larger shifts in the mapping from private to public confidence are required when playing with both low-confidence and high-confidence partners (*Figure 3B*), whereas the quadratic coherence term controls for potential non-linear influences of an internal sense of confidence on the neural response (*Mazor et al., 2020*).

Under this model, we would expect activity in neural areas which track private confidence to vary as a function of motion coherence, whereas neural areas which support a mapping from private to

public confidence should also track information about the social context. Consistent with a role in private-public mappings, FPl tracked both motion coherence and social context (*Figure 3C*; see statistical results in *Table 1*). In particular, FPl activity was higher for higher levels of coherence and for both low-confidence and high-confidence partners – contexts that involved a greater need for the employment of a context-dependent private-public mapping – as indexed by the quadratic context term. As for the ROIs hypothesised to support the formation of an internal sense of confidence, only pgACC tracked motion coherence (*Figure 3C*). While an encoding of motion coherence on its own is insufficient to establish an area as a neural substrate for private confidence, this result is consistent with the previous finding that pgACC tracked a combination of variables that underpinned changes in private confidence (*Bang and Fleming, 2018*).

Finally, we ran an exploratory whole-brain analysis using the same GLM as in the ROI analysis (see Appendix 1). Here we focus on context-related effects: while neural encoding of an internal sense of confidence has been investigated by previous research, our study is the first to manipulate the contextual requirements on the mapping of this internal sense onto explicit confidence reports. We observed activations in relation to the quadratic context term in neural areas that are typically implicated in studies of social cognition – including bilateral temporoparietal junction (*Frith and Frith, 2012*; *Saxe, 2006*) – and cognitive control – including a region anterior to our dACC ROI and in right dorsolateral PFC (*Duncan, 2010*). Notably, and consistent with our ROI analysis, the quadratic context term also revealed significant clusters in bilateral anterior-lateral PFC that overlapped with our FPl ROI (*Figure 3D*).

## Encoding of trial-by-trial confidence in prefrontal cortex

The preceding analysis of neural activity utilised our experimental dissociation of private and public confidence and did not directly incorporate subjects' behaviour. In order to further probe the contribution of our ROIs to a context-dependent mapping from private to public confidence, we next

**Table 1.** Encoding of motion coherence and social context in lateral frontal pole.

Table shows statistical results for the analysis of ROI responses to the context screen shown in *Figure 3C*. The model (GLM1) included separate condition regressors for trials where the context was signalled and trials where the context was hidden. The condition regressor for signalled-context trials was parametrically modulated by linear and quadratic terms for coherence (K and $K^2$) and context (C and $C^2$). In addition to the contrast estimates for these parametric modulators, the table also shows the contrast between signalled-context and hidden-context trials. Statistical testing was performing by comparing contrast estimates across subjects to zero using a one-sample *t*-test. dACC: dorsal anterior cingulate cortex. pgACC: perigenual anterior cingulate cortex. FPl: lateral frontal pole.

| ROI | Contrast | Mean | SEM | t | P |
|---|---|---|---|---|---|
| dACC | K | 0.0646 | 0.1748 | 0.3694 | 0.7147 |
| | $K^2$ | 0.0293 | 0.1492 | 0.1966 | 0.8456 |
| | C | -0.1930 | 0.1503 | -1.2840 | 0.2101 |
| | $C^2$ | 0.3448 | 0.1736 | 1.9857 | 0.0573 |
| | Signalled vs. hidden | 0.6045 | 0.3685 | 1.6402 | 0.1126 |
| pgACC | K | 0.4427 | 0.1616 | 2.7389 | 0.0108 |
| | $K^2$ | 0.0008 | 0.1839 | 0.0043 | 0.9966 |
| | C | 0.2469 | 0.1994 | 1.2384 | 0.2262 |
| | $C^2$ | -0.0878 | 0.2127 | -0.4126 | 0.6831 |
| | Signalled vs. hidden | 0.8477 | 0.4412 | 1.9213 | 0.0653 |
| FPl | K | 0.6132 | 0.1496 | 4.0981 | 0.0003 |
| | $K^2$ | -0.1349 | 0.2314 | -0.5830 | 0.5647 |
| | C | -0.1172 | 0.2118 | -0.5535 | 0.5845 |
| | $C^2$ | 0.9070 | 0.1941 | 4.6741 | 0.0001 |
| | Signalled vs. hidden | 1.2872 | 0.5155 | 2.4967 | 0.0189 |

used subjects' confidence reports to unpack ROI response profiles at a trial-by-trial level. As shown in *Figure 2*, subjects' confidence reports reflect factors relating both to the perceptual decision and the social context. A simple correlation between reported confidence and ROI activity would therefore not reveal whether the relationship was driven by private or public aspects of confidence, or both. To separate the contribution of the perceptual decision and the social context, we leveraged a previously established model-based approach (*Bang and Fleming, 2018*) to estimate the confidence that subjects would have reported on a given trial had there been no contextual modulation – an estimate that could then be compared to the confidence that subjects actually reported.

Our model-based estimate of private confidence is obtained by (1) fitting an ordinal regression model to the behavioural session in order to characterise the influence of motion coherence, choice reaction time and each social context on a subject's confidence reports and (2) applying the fitted model to data from the fMRI session while setting the influence of each social context to zero

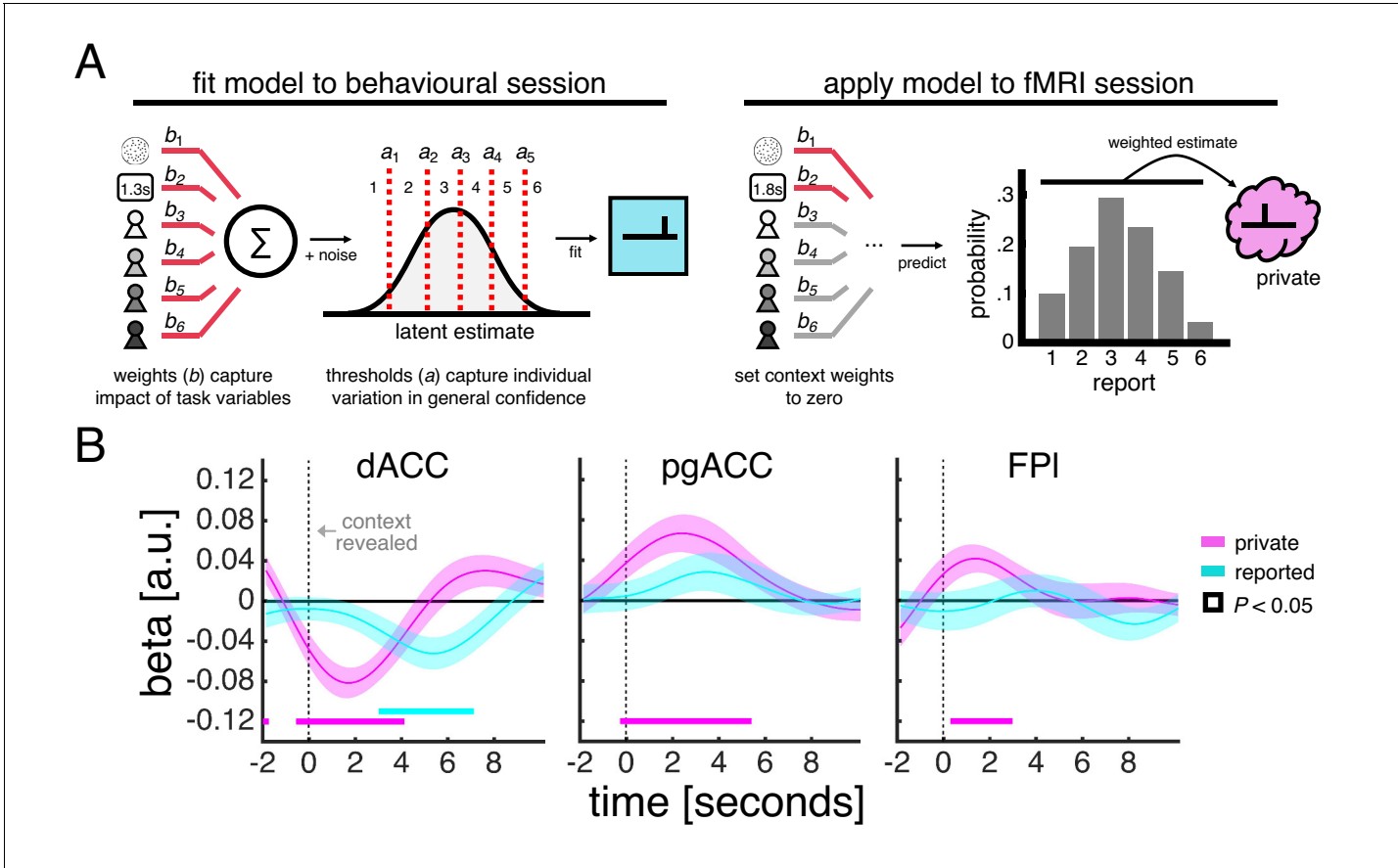

**Figure 4.** Encoding of private and reported confidence in prefrontal cortex. (**A**) Model-based estimate of private confidence. We fitted an ordinal regression model to a subject's confidence reports from the behavioural session (we used the data from the final phase as this phase had the same task design as the fMRI session). The model has a set of weights, which parameterise the influence of the perceptual decision (coherence and choice reaction time) and social context (one term for each partner), and a set of thresholds, which parameterise report biases. We used the fitted model to predict the confidence that a subject would have reported in the fMRI session had there been no contextual modulation – by applying the model to a subject's data while setting the context weights to zero. This prediction is a probability distribution over reports (e.g., a report of '1' has a 10% probability, '2' has 20% probability and so on). We used the expectation under this distribution as our estimate of private confidence. (**B**) GLM analysis of the effects of private confidence and empirically observed confidence reports on ROI activity time courses. Vertical dashed line indicates the onset of the context screen - the context screen, which is presented for 1 s, is shown .5 s after the submission of the perceptual decision and is immediately followed by the confidence scale. Analyses were performed on trials in which the context was explicitly signalled. We tested significance (coloured square) for each time point by comparing coefficients across subjects to zero (p<0.05, one-sample *t*-test).

The online version of this article includes the following figure supplement(s) for figure 4:

**Figure supplement 1.** Model-based estimate of private confidence varies with both motion coherence and choice reaction time.
**Figure supplement 2.** Cross-validation accuracy for confidence model.

(*Figure 4A*). Importantly, this model-based estimate provides a more direct proxy of private confidence than motion coherence alone by also taking into account the time taken to make a decision – a factor which has been shown to affect private confidence over and above the strength of the perceptual evidence (*Kiani et al., 2014*). Indeed, our model-based approach indicated that private confidence was comparable for fast decisions made in response to low-coherence stimuli and slow decisions in response to high-coherence stimuli (*Figure 4—figure supplement 1*).

We related both the model-based estimate of private confidence and the empirically observed confidence reports to ROI activity time courses within a regression framework (*Figure 4B*). Our analysis showed that all three ROIs encoded the model-based estimate of private confidence (pink line). The encoding profiles peaked around 2 s after the onset of the context screen, which, given the slow dynamics of the fMRI signal, is in line with the model-based estimate of private confidence being associated with the earlier perceptual decision rather than the social context. Of the three ROIs, only dACC encoded the empirically observed confidence reports (cyan line). Consistent with a sequential computation of the private and public aspects of confidence, the dACC encoding profile of reported confidence peaked 5–6 s after the onset of the context screen. Given the inclusion of the model-based estimate of private confidence in this analysis, an effect of reported confidence is likely to be driven by the social context. Taken together, these results indicate that dACC tracks the input and the output of a private-public mapping.

We next reasoned that, if FPl provides the context-dependent private-public mapping that is used to transform a private state (input) into a public report (output), then connectivity between FPl and dACC/pgACC should vary with the contextual requirements of the task. In particular, connectivity should be modulated by the relationship between 'what I would have said' had there been no contextual modulation and 'what I actually said'. To test this prediction, we conducted a psychophysiological interaction analysis that quantified connectivity between FPl and dACC/pgACC as a function of the difference between the model-based estimate of private confidence and the empirically observed confidence reports. We included both the individual variables as well as their interaction to allow for differences in the impact of both understating (i.e. private > reported) and overstating (i.e. private < reported) confidence on connectivity. Our analysis revealed a close coupling between FPl and dACC (*Figure 5A*). First, around the onset of the context screen, there was a transient increase in FPl-dACC connectivity associated with the model-based estimate of private confidence (pink line). Second, 6–8 s after the onset of the context screen, FPl-dACC connectivity varied with not only the model-based estimate of private confidence but also the empirically observed confidence reports (cyan line) and their interaction (green line). Visualisation of these effects showed that FPl-dACC connectivity was (1) higher for larger shifts in the mapping from private to reported confidence (off-diagonal elements are warmer than diagonal elements in *Figure 5B*) and (2) highest when confidence was understated rather than overstated (bottom-right elements are warmer than top-left elements in *Figure 5B*). Taken together, these results suggest that dACC integrates contextualised signals from FPl in order to guide trial-by-trial behaviour.

## Representation of task space in lateral frontal pole

Finally, we reasoned that, if FPl indeed orchestrates the context-dependent mapping from private to public confidence, then this area should also carry detailed information about the different social situations engendered by our task. In computational terms, our task comprises 16 states (social situations), with each state corresponding to a combination of coherence and context (e.g., the state on the current trial may be 'low coherence + high-confidence partner', whereas on the next trial a new combination of coherence and context will be encountered). By design, each of these 16 states requires subtly different behavioural responses in order to maximise reward – a relationship reflected by subjects' confidence reports (*Figure 2C*). If FPl supports this contextual modulation of behavioural responses, then it should represent the different states of the task as distinct.

We tested this prediction using representational similarity analysis (RSA) and a metric known as the exemplar discriminability index (EDI) (*Nili et al., 2020*). Like other multivariate methods, RSA considers the pattern of activity across voxels within an ROI rather than the mean activity across voxels. The EDI metric asks whether a multivariate pattern is more stable across scan runs *within* conditions than *between* conditions – the intuition being that higher within-condition stability shows that an ROI represents the conditions as distinct (*Figure 6A*). We obtained the condition-specific multivariate patterns by modelling the neural response to the context screen separately for each

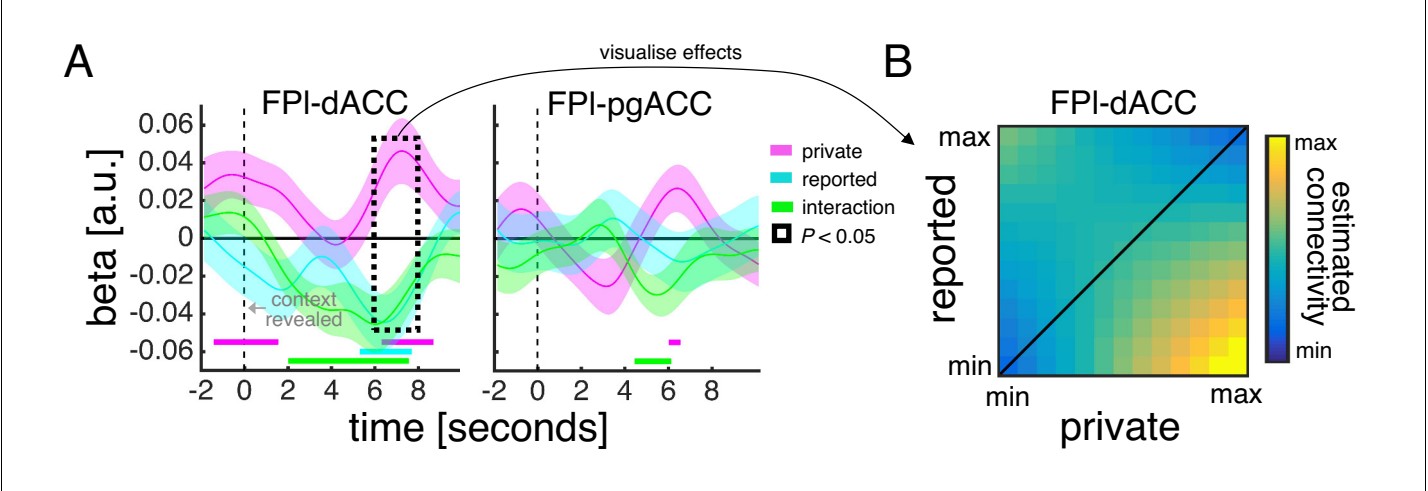

**Figure 5.** Functional connectivity between medial and lateral prefrontal context varies with contextual requirements of task. (**A**) Psychophysiological interaction analysis of ROI activity time courses. Traces are coefficients from a GLM in which we predicted dACC/pgACC activity from the interaction between FPl activity and (1) the model-based estimate of private confidence (pink), (2) the empirically observed confidence reports (cyan) and (3) the interaction between private and reported confidence (green) – while controlling for the main effect of each term. Vertical dashed line indicates the onset of the context screen - the context screen, which is presented for 1 s, is shown .5 s after the submission of the perceptual decision and is immediately followed by the confidence scale. Analyses were performed on trials in which the context was explicitly signalled. We tested significance (coloured square) for each time point by comparing coefficients across subjects to zero (p<0.05, one-sample *t*-test). (**B**) Visualisation of FPl-dACC connectivity. Hotter colours indicate greater FPl-dACC connectivity as a function of variation (in *z*-score units) in private confidence (x-axis) and reported confidence (y-axis). FPl-dACC connectivity was estimated using group-level coefficients averaged across a time window from 6 s to 8 s (box in panel A).

condition of our factorial design (GLM3). In support of a role in orchestrating private-public mappings, FPl – unlike dACC or pgACC – carried a representation of the full task space as well as the sub-spaces of coherence and context (*Figure 6B*; see statistical results in *Table 2*).

## Discussion

Contextual modulation of social behaviour – a core part of healthy social function – is facilitated by the ability to map private states onto public actions in a flexible manner. Here we studied how the brain supports such private-public mappings, by using a social perceptual decision task that required subjects to use learned context-dependent mappings from private to public confidence in order to maximise reward. Combining univariate and multivariate analyses of fMRI data, we found that a private-public distinction is reflected in a medial-lateral division of prefrontal cortex. In line with recent studies (*Bang and Fleming, 2018*; *Fleming et al., 2018*), we found that dACC and pgACC tracked aspects of private confidence – as estimated by a model-based approach that controlled for the impact of the social context on subjects' confidence reports. Further, we found that dACC tracked not only private confidence – the 'input' to a private-public mapping – but also the confidence that subjects actually reported under the contextual requirements of our task – the 'output'. By contrast, FPl, which was hypothesised to govern the mapping from private to public confidence itself, tracked the need for a context-dependent private-public mapping in our task and carried a high-dimensional representation of the different social situations induced by our task. Finally, and in support of a role for FPl in orchestrating the mapping from private to public confidence, FPl-dACC connectivity varied with the contextual requirements of our task – with an increase in connectivity when a larger shift in the mapping from private to public confidence was required.

More broadly, our results are in line with a role of lateral PFC in cognitive control, defined as the ability to use task representations to guide thought and behaviour (*Badre, 2008*; *Badre and Nee, 2018*). In a typical study on cognitive control, the appropriate stimulus-response mapping on a particular trial depends on a contextual cue (e.g., respond '1' when stimulus is blue in context A and '2' in context B). The complexity of the control problem is then increased by introducing additional hierarchical rules (e.g., another cue indicating whether to respond to the colour or size of the stimulus).

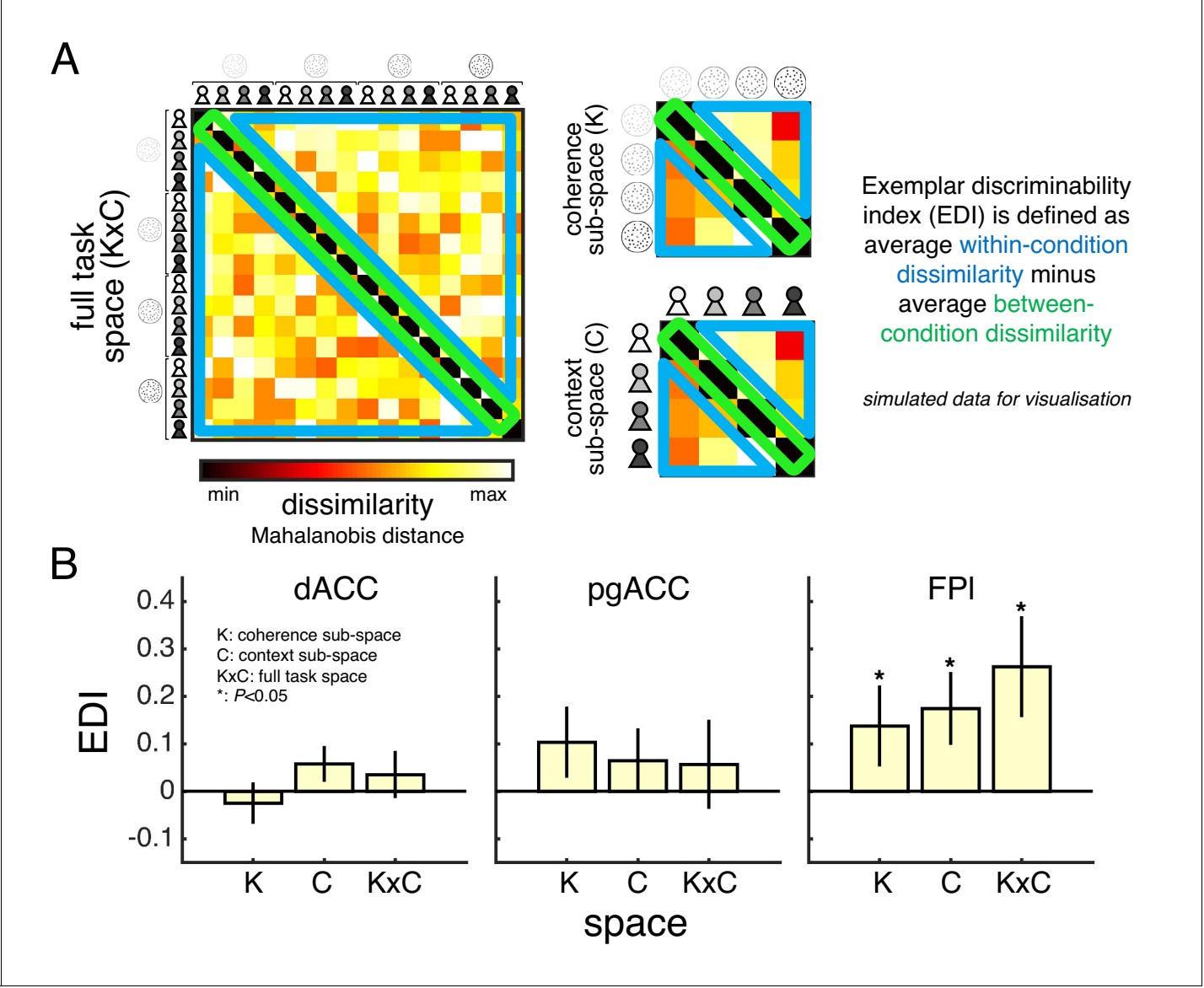

**Figure 6.** Task space representation in lateral frontal pole. (A) A split-data representational dissimilarity matrix (sdRDM) is constructed by (1) computing the Mahalanobis distance between the voxel activity pattern in scan run $i$ and the voxel activity pattern averaged across scan runs $j \neq i$ for every pair of conditions and (2) averaging the sdRDMs across scan runs. The exemplar discriminability index (EDI) is computed as the average dissimilarity across off-diagonal elements (blue) minus the average dissimilarity across diagonal elements (green). A positive EDI indicates that the voxel activity pattern within an ROI is more stable within conditions than between conditions and therefore that the ROI discriminates between the conditions. Simulated data are presented to aid visualisation. (B) ROI EDIs for the full task space (KxC) as well as the sub-spaces of coherence (K) and context (C). We tested significance (asterisk) by comparing EDIs across subjects to zero (p<0.05, one-tailed sign-rank test). Statistical results are summarised in *Table 2*. Data are represented as group mean ± SEM. See Appendix 3 for whole-brain EDI searchlight analysis. dACC: dorsal anterior cingulate cortex. pgACC: perigenual anterior cingulate cortex. FPl: lateral frontal pole.

Both functional neuroimaging and patient studies indicate that lateral PFC is required when control problems increase in complexity – for example, by providing a model of the current task abstracted across individual episodes or acquired through instruction (*Badre and D'Esposito, 2007*; *Badre et al., 2009*). Our results suggest that similar neural and cognitive mechanisms are recruited to solve the private-public mapping problem inherent to social interaction. One open question is, however, whether private-public mappings involve additional computations beyond those invoked by typical cognitive control paradigms, given that the 'stimulus' requiring a different response involves an intervening subjective (metacognitive) state.

**Table 2.** Task space representation in lateral frontal pole.

Table shows statistical results for the analysis of ROI task space representations shown in **Figure 6B**. Condition-specific multivariate patterns were obtained by modelling the neural response to the context screen separately for each condition of our factorial design (GLM3; only signalled-context trials). ROI EDIs were then computed separately for the full task space (KxC) as well as the sub-spaces of coherence (K) and context (C). Statistical testing was performed by comparing EDIs across subjects to zero using a one-tailed sign-rank test. dACC: dorsal anterior cingulate cortex. pgACC: perigenual anterior cingulate cortex. FPl: lateral frontal pole.

| ROI | Space | Mean | SEM | P |
|---|---|---|---|---|
| dACC | K | -0.0249 | 0.0436 | 0.7772 |
| | C | 0.0577 | 0.0379 | 0.1073 |
| | KxC | 0.0353 | 0.0499 | 0.1397 |
| pgACC | K | 0.1033 | 0.0749 | 0.1073 |
| | C | 0.0646 | 0.0678 | 0.2029 |
| | KxC | 0.0567 | 0.0939 | 0.2437 |
| FPl | K | 0.1378 | 0.0854 | 0.0334 |
| | C | 0.1746 | 0.0768 | 0.0258 |
| | KxC | 0.2624 | 0.1062 | 0.0063 |

An alternative explanation of our results is that a division of labour between medial and lateral PFC reflects the engagement of a serial-stage process – a perceptual decision followed by a confidence report – rather than the resolution of a private-public mapping. This explanation is, however, not supported by the data. The quadratic context term – encoded by FPl – compares conditions that are matched in general task characteristics and only differ in the contextual requirements on a private-public mapping. In particular, both 'inlying' contexts (medium-low and medium-high confidence) and 'outlying' contexts (low and high confidence) require sequential preparation of a private state and a public action – the difference between these two types of context is that the latter requires a larger shift in the mapping from private to public confidence. Further, we found that FPl activity was higher on trials where the context was directly signalled than on trials where the context was hidden from subjects (**Table 1**). Again, these two types of trial are matched in general task characteristics and only differ in the availability of a context-dependent private-public mapping. We highlight that these 'matched' comparisons also rule out the alternative explanation that the FPl activations reflect a distinction between implicit and explicit processing (**Shea and Frith, 2016**).

Our results may prompt a re-evaluation of the contribution of lateral PFC to metacognition – the ability to monitor and evaluate our ongoing thought and behaviour. Several studies have shown that lateral PFC tracks explicit reports of confidence (**De Martino et al., 2013**; **Fleming et al., 2012**), and that the microstructure of lateral PFC predicts the degree to which an individual's confidence reports reflect their objective performance (**Allen et al., 2017**; **Fleming et al., 2010**). These results are often taken to show that lateral PFC underpins the computation of an internal sense of confidence, or the 'read-out' of such a variable from circuits involved in decision-making. Our study supports and refines this latter hypothesis, by showing that FPl may contextualise an internal sense of confidence for explicit report in accordance with task requirements. On this account, a relationship between anterior prefrontal structure or function and individual differences in metacognition may not reflect a contribution to insight per se but, rather, the ability to maintain a stable private-public mapping. More broadly, our results fit with a hypothesis that variation in metacognitive biases (e.g., overconfidence) reflect social rather than cognitive factors (**Bang et al., 2017**) – a hypothesis supported by evidence that a private sense of confidence may in fact be computed in a statistically optimal manner (**Aitchison et al., 2015**; **de Gardelle and Mamassian, 2014**; **Sanders et al., 2016**).

Recent years has seen an interest in how the brain encodes structured representations of the world for efficient learning and decision-making (**Behrens et al., 2018**; **Niv, 2019**; **Radulescu et al., 2019**). In these scenarios, context may be particularly important for adaptive behaviour. In psychology, context is defined as the set of circumstances surrounding an event (e.g., we may hear a ringing

phone in different environments such as our home or a friend's house) (*Gershman, 2017*; *Maren et al., 2013*). Representing context is useful because it may carry information about what to expect or what to do (e.g., we should answer a ringing phone at home but not at a friend's house) (*Gershman, 2017*; *Maren et al., 2013*). Context may also hold structural information that can be generalised across events (e.g., many behaviours that are acceptable at home are not acceptable when visiting others) (*Gershman, 2017*). There is evidence that such structural knowledge is encoded in the medial temporal lobe and ventral PFC (*Behrens et al., 2018*; *Niv, 2019*). Our results complement this work by identifying representations in FPl that may support a mapping between private states and public actions that are appropriate for the current context.

Our study highlights several questions for future research. First, how are context-dependent private-public mappings learned? Model-free reinforcement learning (e.g., Q-learning) (*Sutton and Barto, 1998*) may be able to learn the value of taking a public action given a private state and a social context but is likely to be too memory intensive (e.g., storing Q-values for every state-action pair in every context) and too slow (e.g., having to experience every state-action pair in every context) without model-based components (e.g., using a model of the world to generalise across state-action pairs or across contexts based on their similarity) (*Gershman, 2017*). Second, what happens when there is uncertainty about the current context? In our task, the context was either explicitly signalled or fully unknown. However, in everyday life, context typically falls into neither of these categories – it is partially observable and can be inferred from available cues. At a neural level, inference on context is likely to require the involvement of additional areas such as the hippocampus which has been hypothesised to support inference and generalisation based on structural knowledge (*Eichenbaum, 2017a*; *Eichenbaum, 2017b*). Finally, what is the nature of the PFC task representation identified by our study? In our task, each social context is associated with a distinct private-public mapping. It is therefore hard to tell whether PFC discriminates between social contexts or a more abstract construct such as behavioural policies. This question could be addressed by using a version of our task in which seemingly distinct social contexts require subjects to adopt the same private-public mapping.

Here we focused on confidence as a canonical computation that is often the target of private-public mappings, but our approach may be adapted to other internal states in order to further elucidate the neural basis of flexible social behaviour. We achieved contextual shaping of social behaviour through a group decision rule under which subjects had to adapt the mapping from private to public confidence according to the social context in order to maximise reward. However, not all domains involving dissociations between private and public aspects of mental states – such as emotions and preferences – can be readily embedded within a decision task. Nevertheless, incentive structures may be imposed onto the mapping between a private state and a public action in a manner that mimics social life. For example, subjects may be rewarded for understating experimentally controlled feelings of pain in one context but rewarded for overstating them in another one. Similarly, subjects may be rewarded for understating their preferences for consumer items – whose valuation can be established experimentally – in one context but rewarded for overstating them in another one. An open question is the extent to which the PFC activations observed in our study will generalise across domains.

The current conceptualisation of private-public mappings may offer insight into the multiple routes to social dysfunction across a range of neuropsychiatric conditions. Broadly, contextual modulation of social behaviour requires at least three distinct sets of computation – (1) context inference, (2) action selection and (3) learning from outcomes – each presumably supported by distinct neural substrates. In future, it may be possible to distinguish different aspects of social dysfunction along each of these dimensions. For example, context-inappropriate social behaviour may arise because an individual cannot identify the current context (e.g., due to hippocampal damage), cannot inhibit prepotent but context-inappropriate actions (e.g., due to prefrontal damage), or does not make correct inferences from past experience (e.g., due to disturbance in neuromodulatory systems supporting learning). Our study provides a starting point for developing such computational-level characterisations of social dysfunction.

## Materials and methods

### Subject details

Twenty-eight adults (14 females; mean age ± SD, 26.36 ± 7.00) took part in the study. Sample size was determined based on common sample sizes in the field and in order to balance power and resource constraints (*Poldrack, 2019*; *Poldrack et al., 2017*). Each subject received a flat rate for participation (£45, £10/hour) and a performance-based bonus (see below). All subjects provided informed consent including consent to publish and sharing of anonymised data. The study was approved by the Ethics Committee of University College London (8231/001).

### Experimental details

Task and procedure

Subjects performed a social perceptual decision task in separate prescan and scan sessions (*Figure 1—figure supplement 2*). We made modifications to the task within and between these sessions. Here we first describe the full task, before detailing the changes made.

Each trial began with the presentation of a fixation cross at the centre of a circular aperture. After a uniformly sampled delay (prescan:. 5–1 s; scan: 1–4 s), subjects viewed a field of moving dots inside the aperture (.4 s). Once the stimulus terminated, subjects had to press one of two buttons to indicate whether the average direction of dot motion was left or right. Once a choice had been registered, the fixation cross turned grey (.5 s). Subjects were then presented with a screen informing them about their partner on the current trial (1 s). There were four partners in total. Each partner was indicated by a unique colour and name (randomised across subjects). Subjects were told that the partners were created by replaying the responses of people performing the perceptual task on another day but, in reality, all partners were simulated.

Next, subjects had to indicate their confidence in the perceptual decision, by moving a marker along a scale from 1 to 6 in steps of 1. The marker started randomly in one of the six locations on the scale and was controlled by button press. Once a response had been registered, the marker turned grey (.5 s). Subjects were then presented with the partner's response on the corresponding trial (1 s), and the individual decision made with higher confidence was selected by the computer as the group decision, highlighted by a yellow triangle (2 s). Finally, subjects received feedback about the accuracy of the group decision (2 s), before continuing to the next trial. The feedback was indicated by replacing the yellow triangle with a green plus sign if the group decision was correct and a red cross if incorrect. Subjects were incentivised to help each group achieve as many correct decisions as possible: they were told that we would randomly select two trials for each group in each session (4 × 2 × 2 = 16 trials) and pay £1 in bonus for every correct group decision (in reality, all subjects received £10 in bonus).

We varied two features of the task in a factorial (4 × 4) manner. First, we varied the fraction of coherently moving dots (coherence) to manipulate subjects' internal sense of confidence in a perceptual judgement (see stimulus calibration for specification of coherence levels). Second, we specified the partners (context) such that they had the same choice accuracy as subjects but differed in mean confidence (see simulation of partners).

The behavioural session involved four phases. In phase 1, we calibrated the coherence levels so as to achieve target levels of choice accuracy (see stimulus calibration). In phases 2–4, we trained subjects on the social task. In phase 2, subjects were paired with the four partners in a block-wise manner. In particular, there were 4 cycles of blocks of 10 trials per partner (e.g., A-B-C-D-A-B-C-D-A-B-C-D-A-B-C-D; 4 × 4 × 10 = 160 trials). The identity of the current partner was shown before each block of 10 trials. In phase 3, subjects were paired with the four partners in an interleaved manner, with the identity of the current partner only revealed after a perceptual decision had been made (4 × 40 = 160 trials). In phase 4, the group decision and the group outcome, was played out in the background, with the next trial starting after subjects had indicated their confidence. In addition, we introduced a condition where the social context was hidden (5 × 40 = 200 trials). In each phase, the coherence levels were counterbalanced across trials within a social context, so that each coherence level was experienced the same number of times for each partner. To help subjects keep track of the behaviour of each group, they were informed every 40 trials how often their individual

decision had been selected as the group decision for each partner (15 s). The selection statistics were reset after each phase.

The scan session involved four scan runs, using the same task design as in phase 4 of the prescan session. We matched the distribution of conditions (coherence × context) across scan runs in order to facilitate multivariate analysis of the fMRI data, with four repetitions per condition ($4 \times 4 \times 5 = 80$ trials per scan run). The screen informing subjects about how often their individual decision had been selected as the group decision for each partner was shown every 20 trials. The selection statistics were reset after each scan run.

## Random dot kinematograms

Subjects viewed random dot kinematograms (RDKs) contained in a circular white aperture (7 degrees in diameter). Each RDK was made up of three independent sets of dots (each dot was 0.12 degrees in diameter) shown in consecutive frames. Each set of dots were shown for one frame (about 16ms) and then replotted again three frames later (about 50ms). Each time a set of dots was replotted, a subset of the dots, determined by the motion coherence, $k$, was displaced in the direction of motion at a speed of 5 degrees s$^{-1}$, whereas the rest of the dots were displaced at random locations within the aperture. The motion direction was to the left or the right along the horizontal meridian. The dot density was fixed at 30 dots degrees$^{-2}$ s$^{-1}$. To help subjects maintain fixation, a circular region (0.7 degrees in diameter) at the centre of the aperture was kept free of dots. A set of coherence levels, **K**, was identified for each subject in a separate stimulus calibration session.

## Stimulus calibration

Subjects performed the perceptual part of the task in two blocks. In block 1, we deployed a set of prespecified coherence levels, **K**: $\{.03, .06, .12, .24, .48\}$. Each coherence level was used 20 times for each direction (5 x 20 x 2 = 200 trials). We then fitted a simple signal detection theory model with a single noise parameter, $\sigma$, governing the statistical relationship between coherence and choice (see simulation of partners for details on model). We selected the noise parameter which minimised the sum of squared errors between predicted and observed choice accuracy across coherence levels. In block 2, we used the fitted noise parameter to select a set of coherence levels associated with the target choice accuracies 60%, 70%, 80% and 90%, **K**: $\{k_{.6}, k_{.7}, k_{.8}, k_{.9}\}$. Each coherence level was then used 25 times for each direction (4 x 25 x 2 = 200 trials). We repeated the fitting procedure and selected a final set of coherence levels for the main task (the final fitted noise parameter was in turn used to simulate the partners).

## Simulation of partners

We used a signal detection theory model to simulate the partners' choices and confidence reports in phases 2 and 3 of the prescan session (*Bang et al., 2017*). In this model, an agent receives noisy sensory evidence, $x$, sampled from a Gaussian distribution, $x \in N(dk, \sigma)$, and makes a choice by comparing the sensory evidence to zero, choosing left if $x<0$ and right if $x>0$. The mean of the Gaussian distribution is given by coherence, $k$, and direction, $d$, with $d = -1$ indicating left and $d = -1$ indicating right. The standard deviation, $\sigma$, is the level of sensory noise – specified by fitting the model to a subject's choices in the stimulus calibration session (see stimulus calibration). The agent computes an internal estimate of decision confidence, $z$, using the absolute value of the evidence strength, $z = |x|$ – a quantity which is monotonically related to the probability that the perceptual choice is correct given the sensory evidence and the level of sensory noise. Finally, the agent maps this internal estimate onto a confidence report, $r$, by applying a set of thresholds, $r = f(z)$. Precise control over the number of times that the agent selects a particular confidence report is achieved by first simulating a vector of $x$'s – using the known sequence of stimuli in the task – and then setting the thresholds in $z$-space so as to achieve a desired confidence distribution. In this way, we created partners with low, medium-low, medium-high and high mean confidence (see *Figure 1—figure supplement 3* for confidence distributions). We simulated phases 2 and 3 separately, so that a particular partner had the same confidence distribution in each phase. In addition, we simulated $x$'s for each partner in each session under the constraint that their choice accuracy was within 1% of the target choice accuracy for each coherence level (see stimulus calibration). In phase 4, and in the scan session, we did not simulate responses. Instead, to calculate how often a subject's individual decision had been

selected as the group decision, we first computed for each trial the probability of a subject's decision being selected given their confidence report and the partner's confidence distribution and then averaged these probabilities across trials.

## Behavioural analysis

### Regression analysis

We used ordinal regression (probit) to analyse subjects' trial-by-trial confidence reports. The model included (contrast-coded) coherence and context as predictors of interest and (log-transformed) choice reaction time, choice, motion direction and marker starting position as predictors of no interest. We z-scored all variables before analysis. We excluded trials in which the partner's identity was hidden. We performed a separate regression for each subject. We tested the group-level significance of a predictor by comparing the coefficients across subjects to 0 ($p < 0.05$, one-sample $t$-test).

### Confidence model

We used a previously established approach (*Bang and Fleming, 2018*) to construct a model of private confidence for fMRI analysis. We fitted an ordinal regression model to a subject's confidence reports in the final phase of the behavioural session using six predictors: (1) z-scored, contrast-coded coherence, (2) z-scored, log-transformed choice reaction time and (3-6) a dummy variable for each explicitly signalled context. We then applied the fitted model to a subject's data in the fMRI session, while setting the fitted context coefficients to zero. This approach yields an out-of-sample prediction about the level of confidence that a subject would have reported on a given trial in the absence of contextual (i.e. partner-specific) modulation. The prediction takes the form of a probability distribution over possible responses (e.g., a report of '1' has a 10% probability, '2' has a 20% probability and so on). We used the expectation under this distribution as our estimate of private confidence.

## fMRI analysis

### Acquisition

MRI data were acquired on a 3T Siemens Prisma scanner with a 64-channel head coil. T1-weighted structural images were acquired using a 3D MPRAGE sequence: $1 \times 1 \times 1$ mm resolution voxels; 176 sagittal slices, $256 \times 224$ matrix; TR = 2530 ms; TE = 3.34 ms; TI = 1100 ms. BOLD T2*-weighted functional images were acquired using a gradient-echo EPI pulse sequence: $3 \times 3 \times 3$ mm resolution voxels; 48 transverse slices, $64 \times 74$ matrix; TR = 3.36; TE = 30 ms; slice tilt = 0 degrees, slice thickness = 2 mm; inter-slice gap = 1 mm; ascending slice order. Field maps were acquired using a double-echo FLASH (gradient echo) sequence: TE1 = 10 ms; TE2 = 12.46 ms; 64 slices were acquired with 2 mm slice thickness and a 1 mm gap; in-plane field of view is $192 \times 192$ mm$^2$ with $3 \times 3$ mm$^2$ resolution.

### Preprocessing

MRI data were pre-processed using SPM12. The first 4 volumes of each functional run were discarded to allow for T1 equilibration. Functional images were slice-time corrected, realigned and unwarped using the field maps (*Andersson et al., 2001*). Structural T1-weighted images were co-registered to the mean functional image of each subject using the iterative mutual-information algorithm. Each subject's structural image was segmented into grey matter, white matter and cerebral spinal fluid using a nonlinear deformation field to map it onto a template tissue probability map (*Ashburner and Friston, 2005*). These deformations were applied to structural and functional images to create new images spatially normalised to the Montreal Neurological Institute (MNI) space and interpolated to $2 \times 2 \times 2$ mm voxels. Normalized images were spatially smoothed using a Gaussian kernel with full-width half-maximum of 8 mm. The motion correction parameters estimated from the realignment procedure and their first temporal derivatives – 12 'motion' regressors in total – were included as confounds in the first-level analysis for each subject.

### Physiological monitoring

Peripheral measurements of a subject's pulse and breathing were made together with scanner slice synchronisation pulses using a Spike2 data acquisition system (Cambridge Electronic Design Limited, Cambridge UK). The cardiac pulse signal was measured using an MRI compatible pulse oximeter

(Model 8600 F0, Nonin Medical, Inc Plymouth, MN) attached to a subject's finger. The respiratory signal, thoracic movement, was monitored using a pneumatic belt positioned around the abdomen close to the diaphragm. A physiological noise model was constructed to account for artifacts related to cardiac and respiratory phase and changes in respiratory volume using an in-house MATLAB toolbox (*Hutton et al., 2011*). Models for cardiac and respiratory phase and their aliased harmonics were based on RETROICOR (*Glover et al., 2000*) and a similar, earlier method (*Josephs et al., 1997*). Basis sets of sine and cosine Fourier series components extending to the 3rd harmonic were used to model physiological fluctuations. Additional terms were included to model changes in respiratory volume (*Birn et al., 2006*; *Birn et al., 2008*) and heart rate (*Chang and Glover, 2009*). This procedure yielded a total of 14 'biophysical' regressors that were sampled at a reference slice in each image volume. The regressors were included as confounds in the first-level analysis for each subject.

## Regions of interest

We focused on three a priori ROIs highlighted by previous research on decision confidence. The dACC mask was an 8 mm sphere around the peak coordinates (MNI coordinates [$x$ $y$ $z$] = [0 17 46]) identified by *Fleming et al., 2012*. The pgACC mask was defined using the coherence x distance second-level $t$-map from *Bang and Fleming, 2018*. The FPl mask was defined using the right-hemisphere atlas developed by *Neubert et al., 2014* and mirrored to the left hemisphere to create a bilateral mask.

## Univariate analysis

Univariate analysis of fMRI data was performed using SPM12. Our main analysis was based on an event-related GLM (GLM1) of the neural response to the context screen. This model included three condition regressors. First, the context screen when the partner was signalled (signalled, 1 s boxcar). Second, the context screen when the partner was hidden (hidden, 1 s boxcar). Third, the update screen informing subjects how often their individual decision had been selected as the group decision for each partner (update, 15 s boxcar). We parametrically modulated the signalled condition regressor using our task factors: (1) K, contrast-coded coherence, {−1.5,−0.5,.5,1.5}; (2) K$^2$, contrast-coded coherence squared; (3) C, contrast-coded context, {−1.5,−0.5,.5,1.5}; and (4) C$^2$, contrast-coded context squared. For comparability with earlier studies on decision confidence, we also estimated an event-related GLM (GLM2) of the neural response to the presentation of the motion stimulus. This modelled included one condition regressor – a boxcar lasting the duration of the stimulus (.4 s) – parametrically modulated by linear and quadratic coherence terms as defined above.

Parametric modulators were not orthogonalized. We excluded trials in which subjects' choice reaction times were 2.5 SD below or above their grand mean reaction time within a scan run (0–2 trials per subject per run). In addition to the condition regressors, we added motion and biophysical parameters as additional 'nuisance' regressors. Regressors were convolved with a canonical hemodynamic response function. Regressors were modelled separately for each scan run, and constants were included to account for differences in mean activation between runs and scanner drifts. A high-pass filter (128 s cutoff) was applied to remove low-frequency drifts. Whole-brain statistical testing was performed by applying one-sample $t$-tests against 0 to the first-level contrast images. We report clusters significant at p<0.05, FWE-corrected for multiple comparisons, with a cluster-defining threshold of p<0.001, uncorrected. For the ROI analysis, we extracted mean contrast estimates within the ROI masks from first-level contrast images and assessed group-level significance by applying one-sample $t$-tests against zero (p<0.05) to the extracted contrast estimates.

## Activity time course analysis

We used activity time courses to study the neural encoding of trial-by-trial confidence and assess functional coupling between ROIs. We transformed each ROI mask from MNI to native space and extracted preprocessed BOLD time courses as the average of voxels within each mask. For each scan run, we regressed out variation due to head motion, applied a high-pass filter (128 s cut-off) to remove low-frequency drifts, and oversampled the BOLD time course by a factor of ~23 (time resolution of. 144 s). For each trial, we extracted activity estimates in a 12 s window time-locked to our event of interest (i.e. from 2 s prior to the onset of the context screen to 10 s after its onset). We

excluded trials in which subjects' choice reaction times were 2.5 SD below or above their grand mean reaction time across trials (1–13 trials per subject). We applied a linear regression to each time point and then, by concatenating beta-weights across time points, generated a time course for each predictor of the regression model. We performed a separate analysis for each subject. We tested the group-level significance of a time point by comparing the beta-weights across subjects to 0 (p<0.05, one-sample t-test).

## Multivariate analysis

Representational similarity analysis (RSA) of fMRI data was performed using SPM12 and the RSA toolbox (*Nili et al., 2014*). To estimate voxel activity patterns, we constructed an event-related GLM with a condition regressor locked to the context screen (1 s boxcar) for each coherence × (signalled) context condition (4 × 4 = 16 regressors per scan run). The GLM for ROI analysis (GLM3) was based on unsmoothed data, whereas the GLM for searchlight analysis (GLM4) was based on smoothed data. As in GLM1, we included regressors for hidden context and update screen and motion and biophysiological regressors as additional 'nuisance' variables. Regressors were convolved with a canonical hemodynamic response function. Regressors were modelled separately for each scan run, and constants were included to account for differences in mean activation between runs and scanner drifts. A high-pass filter (128 s cutoff) was applied to remove low-frequency drifts.

The exemplar discriminability index (EDI) is a test of whether an ROI carries information about a set of conditions (*Nili et al., 2020*) – in our case the 16 coherence × context conditions that make up the full task space. The hypothesis is that, if a neural area represents the conditions as distinct, then its voxel activity pattern should be more stable across scan runs *within* conditions than *between* conditions. This hypothesis is tested by first constructing a split-data representational dissimilarity matrix (sdRDM) for each scan run that contains the Mahalanobis distance (i.e. multivariate noise normalisation and Euclidean distance) between the voxel activity pattern in scan run $i$ and the voxel activity pattern averaged across scan runs $j \neq i$ for all condition pairs and then averaging the run-specific sdRDMs across scan runs. An EDI metric is then be computed as the average dissimilarity across the off-diagonal elements minus the average dissimilarity across the diagonal elements (*Figure 6A*). A positive EDI shows that the voxel activity pattern in an area is more stable within conditions than between conditions and therefore that the area carries information about the conditions.

ROI analysis was performed by computing the EDI metric from voxel activity estimates under the first-level model within our ROI masks. As the EDI should theoretically not be below zero, we assessed group-level significance for ROI analysis using a one-tailed sign-rank test (p<0.05). Searchlight analysis – as executed by the RSA toolbox – was performed by computing the EDI metric for a spherical cluster of voxels centred at each voxel within each subject's first-level map. Consistent with previous RSA studies (*Nili et al., 2014*), the diameter of the sphere was 15 mm (around 100 voxels). The sphere was adapted in shape if it was near the edges of a whole-brain group-level mask – the mask was defined as the intersection of the whole-brain masks obtained for each subject during estimation of the first-level model. Whole-brain statistical testing was performed by applying one-sample t-tests against 0 to the first-level EDI maps. We report clusters significant at p<0.05, FWE-corrected for multiple comparisons, with a cluster-defining threshold of p<0.001, uncorrected. We note that the searchlight analysis is not directly comparable to the ROI analysis but, rather, complements it. First, the FPl ROI analysis is more sensitive to subtle pattern information as the FPl ROI contains around 10 times more voxels than the searchlight sphere. Second, our FPl ROI is bilateral and thus contains a substantial proportion of non-contiguous voxels. By contrast, the searchlight sphere is restricted to contiguous voxels within a relatively restricted region and can therefore only identify localised pattern information. Finally, our FPl ROI is anatomically defined and thus has a shape which cannot be captured by a searchlight sphere.

We note that the number of trials per condition per scan run (i.e. 4) is compatible with previous studies employing RSA (*Kriegeskorte et al., 2008*). Constraints on per-subject scanning time means that there is an inherent trade-off between the ability to estimate a neural pattern within a condition (the estimate improves with repetitions of a condition) and the ability to estimate neural pattern dissimilarities between conditions (the estimate improves with the number of conditions). We adopted a condition-rich design as our goal was to test whether an ROI treats the different conditions (i.e. task states) as distinct and not to characterise the ROI pattern for any particular condition. We note

that our analysis of the sub-spaces of coherence and context were based on 16 trials per condition per run and not four trials per condition per run as in the analysis of the full task space.

## Data and code availablity

Data and code for reproducing figures as well as associated analyses are available on GitHub: *Bang, 2020*; https://github.com/danbang/article-private-public (copy archived at https://github.com/elifesciences-publications/article-private-public). Whole-brain group-level statistical maps are available on NeuroVault: https://neurovault.org/collections/6782/.

## Acknowledgements

The Wellcome Centre for Human Neuroimaging is supported by core funding from Wellcome (203147/Z/16/Z). DB is supported by a Sir Henry Wellcome Postdoctoral Fellowship funded by Wellcome (213630/Z/18/Z). SMF is supported by a Sir Henry Dale Fellowship jointly funded by Wellcome and the Royal Society (206648/Z/17/Z).

## Additional information

### Funding

| Funder | Grant reference number | Author |
| --- | --- | --- |
| Wellcome | 213630/Z/18/Z | Dan Bang |
| Wellcome | 206648/Z/17/Z | Stephen M Fleming |
| Royal Society | 206648/Z/17/Z | Stephen M Fleming |

The funders had no role in study design, data collection and interpretation, or the decision to submit the work for publication.

### Author contributions

Dan Bang, Conceptualization, Resources, Data curation, Software, Formal analysis, Funding acquisition, Investigation, Visualization, Methodology, Writing - original draft, Project administration, Writing - review and editing; Sara Ershadmanesh, Investigation, Methodology, Writing - review and editing; Hamed Nili, Conceptualization, Supervision, Methodology, Writing - review and editing; Stephen M Fleming, Conceptualization, Resources, Supervision, Funding acquisition, Methodology, Writing - original draft, Writing - review and editing

### Author ORCIDs

Dan Bang (iD) https://orcid.org/0000-0001-7446-7090
Stephen M Fleming (iD) https://orcid.org/0000-0003-0233-4891

### Ethics

Human subjects: All subjects provided informed consent including consent to publish and sharing of anonymised data. The study was approved by the Ethics Committee of University College London (8231/001).

### Decision letter and Author response

Decision letter https://doi.org/10.7554/eLife.56477.sa1
Author response https://doi.org/10.7554/eLife.56477.sa2

## Additional files

### Supplementary files

• Transparent reporting form

## Data availability

Data and code for reproducing figures as well as associated analyses are available on GitHub: https://github.com/danbang/article-private-public (copy archived at https://github.com/elifescien-ces-publications/article-private-public). Whole-brain group-level statistical maps are available on NeuroVault: https://neurovault.org/collections/6782/.

The following dataset was generated:

| Author(s) | Year | Dataset title | Dataset URL | Database and Identifier |
|---|---|---|---|---|
| Bang D, Ershad-manesh S, Nili H, Fleming SM | 2020 | Private-public mappings in human prefrontal cortex | https://neurovault.org/collections/6782/ | NeuroVault, 6782 |

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

## Appendix 1

Whole-brain activations in response to context screen. The model (GLM1) included separate 'condition' regressors for trials where the context was signalled and trials where the context was hidden. The condition regressor for signalled-context trials was parametrically modulated by linear and quadratic terms for coherence (K and $K^2$) and context (C and $C^2$). In addition to the contrast estimates for these parametric modulators, the table also shows the contrast between signalled-context and hidden-context trials. Whole-brain statistical testing was performed by comparing contrast images across subjects to zero using one-sample $t$-tests. All reported activations are significant at $p<0.05$, FWE-corrected for multiple comparisons, with a cluster-defining threshold of $p<0.001$, uncorrected. Only clusters with a minimum size of 100 voxels are reported. K: coherence. C: context. FWE: familywise error. MNI: Montreal Neurological Institute. L: left. R: right. TPJ: temporoparietal junction. dmPFC: dorsomedial prefrontal cortex. FPl: lateral frontal pole.

| Contrast | Label | voxels at p<0.001 | Peak z-score | P (cluster FWE corrected) | Peak Voxel MNI coordinates | | | |
|---|---|---|---|---|---|---|---|---|
| K: positive | striatum | 1021 | 5.48 | <0.001 | -20 | 8 | 10 | L |
| | striatum | 852 | 5.18 | <0.001 | 18 | 14 | 0 | R |
| | posterior cingulate | 419 | 4.49 | <0.001 | -4 | -34 | 32 | LR |
| | cerebellum | 479 | 4.45 | <0.001 | -28 | -62 | -40 | L |
| | cerebellum | 281 | 4.41 | 0.003 | 22 | -72 | -28 | R |
| | parietal | 196 | 4.30 | 0.019 | -36 | -60 | 62 | L |
| | parietal | 462 | 4.11 | <0.001 | 42 | -52 | 44 | R |
| K: negative | | | | | | | | |
| $K^2$: positive | | | | | | | | |
| $K^2$: negative | | | | | | | | |
| C: positive | occipital | 222 | 4.23 | 0.015 | 20 | -80 | -10 | R |
| C: negative | occipital | 277 | 3.71 | 0.005 | -6 | -84 | -14 | L |
| $C^2$: positive | parietal (incl. TPJ) | 1668 | 5.57 | <0.001 | -40 | -58 | 36 | L |
| | parietal (incl. TPJ) | 1656 | 4.82 | <0.001 | 50 | -42 | 44 | R |
| | lateral frontal (incl. FPl) | 465 | 4.49 | <0.001 | -38 | 56 | 8 | L |
| | posterior cingulate | 388 | 4.44 | 0.001 | -2 | -24 | 34 | LR |
| | dorsal medial frontal (incl. dmPFC) | 692 | 4.29 | <0.001 | 4 | 36 | 32 | LR |
| | dorsolateral frontal | 330 | 4.25 | 0.002 | 36 | 32 | 40 | R |
| | lateral frontal (incl. FPl) | 598 | 4.05 | <0.001 | 26 | 60 | -8 | R |
| | precuneus | 366 | 3.90 | 0.001 | 10 | -70 | 44 | R |
| $C^2$: negative | amygdala | 196 | 4.54 | 0.029 | 18 | -8 | -20 | R |

*continued*

| Contrast | Label | voxels at p<0.001 | Peak z-score | *P* (cluster FWE corrected) | Peak Voxel MNI coordinates | | | |
|---|---|---|---|---|---|---|---|---|
| signalled > hidden | dorsal medial frontal (incl. dmPFC) | 341 | 4.67 | .001 | 2 | 32 | 32 | LR |
| | posterior cingulate | 1020 | 4.64 | <0.001 | 0 | -14 | 32 | LR |
| | lateral frontal (incl. FPl) | 181 | 4.05 | .035 | 38 | 52 | 14 | R |
| | anterior frontal | 283 | 3.95 | .004 | 6 | 56 | 0 | LR |
| signalled < hidden | temporal lobe | 519 | 4.59 | <0.001 | -68 | -42 | 16 | L |
| | anterior temporal lobe | 264 | 4.17 | .006 | -52 | -2 | -14 | L |
| | temporal lobe | 584 | 4.12 | <0.001 | 60 | -32 | 2 | R |
| | anterior temporal lobe | 220 | 3.87 | .015 | 52 | -12 | -20 | R |

## Appendix 2

Whole-brain activations in response to stimulus presentation. The model (GLM2) included a 'condition' regressor for stimulus presentation. The condition regressor was parametrically modulated by linear and quadratic coherence (K and $K^2$). Whole-brain statistical testing was performed by comparing contrast images across subjects to zero using one-sample $t$-tests. All reported activations are significant at p<0.05, FWE-corrected for multiple comparisons, with a cluster-defining threshold of p<0.001, uncorrected. Only clusters with a minimum size of 100 voxels are reported. K: coherence. FWE: familywise error. MNI: Montreal Neurological Institute. L: left. R: right. IPS: intraparietal sulcus.

| Contrast | Label | voxels at p<0.001 | Peak z-score | P (cluster FWE corrected) | Peak Voxel MNI coordinates | | | |
|---|---|---|---|---|---|---|---|---|
| K: positive | striatum | 241 | 4.41 | .006 | -20 | 2 | 12 | L |
| | striatum | 160 | 4.31 | .042 | 22 | -2 | 0 | R |
| | parietal (incl. IPS) | 821 | 4.25 | <0.001 | 40 | -52 | 38 | R |
| | parietal (incl. IPS) | 506 | 4.08 | <0.001 | -42 | -56 | 46 | L |
| K: negative | | | | | | | | |
| $K^2$: positive | | | | | | | | |
| $C^2$: negative | | | | | | | | |

## Appendix 3

Whole-brain activations in EDI searchlight analysis. Condition-specific multivariate patterns were obtained by modelling the neural response to the context screen separately for each condition of our factorial design (GLM4; only signalled-context trials). An EDI for the full task space (KxC) was then computed for a spherical cluster of voxels centred at each voxel within each subject using a searchlight procedure. Whole-brain statistical testing was performed by comparing EDI images across subjects to zero using one-sample *t*-tests. All reported activations are significant at p<0.05, FWE-corrected for multiple comparisons, with a cluster-defining threshold of p<0.001, uncorrected. Only clusters with a minimum size of 100 voxels are reported. FWE: familywise error. KxC: coherence x context. MNI: Montreal Neurological Institute. L: left. R: right.

| Space | Label | voxels at p<0.001 | Peak *z*-score | *P* (cluster FWE corrected) | Peak Voxel MNI coordinates | | | |
|---|---|---|---|---|---|---|---|---|
| KxC | visual cortex | 12170 | 7.01 | <0.001 | -18 | -86 | -14 | LR |
| | posterior parietal | 791 | 5.12 | <0.001 | -42 | -48 | 36 | R |
| | visual cortex | 141 | 5.10 | <0.001 | 46 | -52 | -6 | R |
| | parietal | 170 | 4.29 | <0.001 | -24 | -24 | 78 | L |

