## [Decision Letter]

**Acceptance summary:**

How individuals convert private mental states into context-dependent public reports is an important but open question. This study tackles this issue in an elegant and novel way, using functional imaging and perceptual reports as an example. The results show that lateral frontal pole supports the mapping of private information to public reports, providing novel insights into the brain areas supporting social behavior.

**Decision letter after peer review:**

Thank you for submitting your article "Private-public mappings in human prefrontal cortex" for consideration by *eLife*. Your article has been reviewed by three peer reviewers, one of whom is a member of our Board of Reviewing Editors, and the evaluation has been overseen by Michael Frank as the Senior Editor. The following individual involved in review of your submission has agreed to reveal their identity: Steve W C Chang (Reviewer #3).

The reviewers have discussed the reviews with one another and the Reviewing Editor has drafted this decision to help you prepare a revised submission. In recognition of the fact that revisions may take longer than the two months we typically allow, until the research enterprise restarts in full, we will give authors as much time as they need to submit revised manuscripts.

Summary:

This study examines context-dependent mapping from "private" to "public" confidence in the human brain using combined univariate and multivariate approaches to fMRI data. The authors used a social perceptual decision task with 4x4 factorial design, which enabled the separation of an internal sense of confidence from an explicit public report. The authors found that subjects considered the social context for making their public confidence ratings. Imaging data was analyzed in medial-lateral divisions of the prefrontal cortex, motivated by previous studies. Univariate fMRI analyses show that pgACC and dACC tracked the formation of an internal sense of confidence, while FPl is involved in mapping it to a public report that reflects the contextual requirement. Finally, multivariate analysis using RSA further support the idea that FPI is involved in contextualizing private confidence by carrying a representation of different state conditions in the task space.

How the brain translates private information into the public domain is an important question, and all reviewers agreed that this study makes an important contribution to this topic. There were also a number of concerns that the authors should address in a revised version of this manuscript.

Essential revisions:

1) There were multiple comments regarding the analyses and results surrounding Figures 3 and 4. These need to be clarified.

1.1) The choice of modeling BOLD for Figure 3 only when the player was revealed seems arbitrary. Other studies looking at "internal" confidence ratings have typically looked at neural signals locked on the type I response onset. Please report the results of modeling BOLD at the response onset of the private and public decision with the same parametric regressors.

1.2) Relatedly, why does pgACC and dACC track the formation of an internal sense of confidence (although each showed the response profile of encoding different measures of it) only after being presented with the social context, but not at the time of making an individual decision (no effects of private confidence seemed to be present before the revelation of the context in Figure 4B)? Is this what the authors expected or is this driven by not having any task marker associated with the time when subjects internally decided on a response?

1.3) The authors report areas that track subjective confidence as well as those that may be involved in private-to-public mapping, but it is unclear which (if any) areas tracks the variable that is ultimately being reported? Figure 4 suggests that dACC activity would be a good candidate. Does dACC correlate with publicly reported confidence (more so than subjective confidence) if an analytical approach akin to Figure 3 is used?

1.4) For all parametric fMRI analyses in Figure 3, please compare the goodness of fit between the linear and quadratic models, to assess the necessity of adding a quadratic expansion. For the conclusions to be warranted, a quadratic term should improve goodness of fit in FPI but not other regions.

1.5) "an indicator of the degree of behavioral deviation from a default policy" – If this statement were true, shouldn't the connectivity between FPl and dACC correlate with the absolute value of the difference between private and public confidence, reflecting the same rationale for using the quadratic context term as parametric regressor?

1.6) There were no explanations for why there might be two significant time periods of PPI effects in Figure 4C. There are to two independent peaks, one right after the context revelation and the other, a more pronounced peak on average, occurring from 6 sec and 8 sec. Please explain these patterns, which are also seen even for the FPI-pgACC panel.

2) Reviewers wondered if the contrast between internal and explicit confidence is contaminated by the choice of time-locking event. That is, are the results really specific to private-to-public mapping, or do they rather reflect the sequential nature of this task, and comparable findings would be observed in a number of processes involving two different states? This question has implications for the novelty of the current study. That is, if the results are not specific for private-public mapping, as opposed to a serial stage, implicit-explicit, etc. type of processes, then the novelty of the neural data would decrease profoundly (e.g., the idea of internal processing to be more medially and external processing to be more laterally localized in the PFC has been around for quite some time).

3) Confidence model.

3.1) It would be important to add a description of the model leading to confidence estimates irrespective of context, as it seems rather mysterious in the present form. Regarding the analysis strategy, please compare this modeling approach with a simpler strategy, e.g., applying mixed-effects ordinal regression on confidence with perceptual evidence and context as fixed effects.

3.2) In order to make sure that the confidence model and its out-of-sample prediction of private confidence estimates are reliable, it would be necessary to validate the accuracy of the model within the data it fitted to. Please report cross-validation accuracy within the data from the behavioral session.

3.3) What is the correlation between motion coherence and the model-derived estimates of private confidence? It was unclear why dACC did not encode the linear term for coherence as in Figure 3C but did track the model-derived estimates of private confidence, although both are the indirect measures of private confidence. Please explain what each of these measures differentially represent with regards to the subject's private state. Also, it was not clear why dACC would only encode one of private confidence measures.

4) Reviewers were concerned that because the current study addresses a novel question with a novel approach and there was no pre-registration, that there was quite a bit of analytical freedom, for instance in the selection of ROIs, which might have biased results. To help mitigate these concerns, it would be important to (a) clearly state the a priori rational that went into the selection of ROIs, (b) fully report the results of the analyses regarding different time points (essential revision 1.1), and (c) show that the results of the ROI analyses converge with the whole-brain results. In this regard, please move the whole-brain univariate results to the main text and discuss it more thoroughly. For the RSA analysis, please conduct a searchlight to provide whole-brain results that can be compared with the ROI findings as well.

5) The manuscript does not contain any statistical results in the main text to support the claims. Some statistical thresholds are reported in the figure legends, but some figures (for instance Figures 4 and 5) do not contain any statistical information. Please include all statistical tests, t, f, and exact p-values in the main text.

---

## [Author Response]

Essential revisions:1) There were multiple comments regarding the analyses and results surrounding Figure 3 and 4. These need to be clarified.1.1) The choice of modeling BOLD for Figure 3 only when the player was revealed seems arbitrary. Other studies looking at "internal" confidence ratings have typically looked at neural signals locked on the type I response onset. Please report the results of modeling BOLD at the response onset of the private and public decision with the same parametric regressors.

We thank the reviewers for this prompt to clarify our key analyses. The focus of the current study is the generation of a context-dependent confidence report – a process that requires the combination of an internal sense of confidence with knowledge about the current context. We selected the context screen as our time window of interest for two reasons. First, the context screen is the first point in a trial that information about the current partner is revealed – any context-related regressors would have little meaning if assigned to timepoints earlier on in a trial. Second, during the presentation of the context screen, subjects have all the information needed to internally decide on a context-dependent confidence report, but the neural response will not be confounded by the engagement of motor processes needed to select a confidence report – a motor plan can only be prepared after the randomised initial location of the confidence marker is revealed. We now carefully describe the reasoning behind the selection of our time window of interest in the main text (subsection “Encoding of motion coherence and social context in prefrontal cortex”).

Nonetheless, we appreciate that any inference on timing here is going to be imprecise due to the lag in the BOLD signal and the short, fixed intervals between the context screen and the preceding perceptual decision (0.5s) and subsequent confidence scale (presented immediately after the context screen). Thus, in line with the reviewers’ suggestion, we now also perform a whole-brain analysis of the BOLD response to the presentation of the motion stimulus using linear and quadratic coherence as parametric modulators (Appendix 2 – statistical maps uploaded to our NeuroVault repository). Note that, as described above, it does not make sense to include regressors related to the context at this timepoint, as the information has not yet been revealed. Further, we reiterate the short, fixed intervals between the context screen and (1) the preceding perceptual decision and (2) the subsequent confidence scale in the figures that display ROI activity time courses (Figure 4 and 5) - thus encouraging the reader to independently evaluate the temporal profile of our results.

1.2) Relatedly, why does pgACC and dACC track the formation of an internal sense of confidence (although each showed the response profile of encoding different measures of it) only after being presented with the social context, but not at the time of making an individual decision (no effects of private confidence seemed to be present before the revelation of the context in Figure 4B)? Is this what the authors expected or is this driven by not having any task marker associated with the time when subjects internally decided on a response?

We thank the reviewers for prompting further clarity on this analysis. We regret that we did not provide sufficient detail in the main text for appreciating (a) the difference between the factorial analysis of neural activity (Figure 3) and the model-based analysis of neural encoding (old Figure 4) and (b) the temporal features of the encoding profiles revealed by our model-based approach. In response to this point – and essential revisions 1.3, 1.6 and 3 – we have substantially revised the section on neural encoding (subsection “Encoding of trial-by-trial confidence in prefrontal cortex”) – including the description of the confidence model – and the associated figure (now divided into Figure 4 and Figure 5).

There are two aspects to point 1.2: (a) possible reasons why the factorial and model-based analyses generate different (but in fact complementary) results and (b) temporal features of the neural encoding of the model-based estimate of private confidence. As for issue (a), our model-based estimate of private confidence is not directly equivalent to the coherence terms in our factorial analysis. Instead, the model-based estimate takes into account not only the influence of motion coherence on subjective experience but also the time taken to make a decision – a factor known to affect private confidence over and above the perceptual evidence itself (Kiani, Corthell and Shadlen, Neuron, 2014). Indeed, our model-based approach indicates that private confidence is comparable for fast decisions in response to low-coherence stimuli and slow decisions in response to high-coherence stimuli (now illustrated in Figure 4—figure supplement 1). As for issue (b), peak neural encoding of the model-based estimate of private confidence appears around 2 seconds after the onset of the context screen – a result which, given the canonical lag in the BOLD signal and the trial timings detailed in response to essential revision 1.1, is consistent with the model-based estimate relating to the perceptual decision rather than the social context. We now explain these relationships carefully in the main text.

1.3) The authors report areas that track subjective confidence as well as those that may be involved in private-to-public mapping, but it is unclear which (if any) areas tracks the variable that is ultimately being reported? Figure 4 suggests that dACC activity would be a good candidate. Does dACC correlate with publicly reported confidence (more so than subjective confidence) if an analytical approach akin to Figure 3 is used?

We recognise that this result may not have been apparent from the section on neural encoding and we have – as described in response to essential revision 1.2 – now revised it significantly. Critically, our analysis of neural encoding includes not only the model-based estimate of private confidence but also the empirically observed confidence reports (pink and cyan lines in Figure 4). Of the three ROIs, only dACC encoded both variables, indicating that dACC indeed covaries with the variable that is ultimately being reported by subjects. We interpret this analysis as indicating that dACC encodes elements of both the “input” and the “output” of a private-public mapping – with our other analyses indicating that FPl provides the relevant signals needed to adjust the mapping itself. We have also included a new opening paragraph of the Discussion to ensure that the findings regarding the three ROIs are clearly summarised.

1.4) For all parametric fMRI analyses in Figure 3, please compare the goodness of fit between the linear and quadratic models, to assess the necessity of adding a quadratic expansion. For the conclusions to be warranted, a quadratic term should improve goodness of fit in FPI but not other regions.

We thank the reviewers for this suggestion. We have added Figure 3—figure supplement 1 where we show – in an independent set of analyses – that (1) a full model (both linear and quadratic terms) provides a better fit to FPl activity estimates than a reduced model (linear terms only) and (2) that the difference in goodness-of-fit between the full and the reduced model is higher in FPl than in other ROIs.

1.5) "an indicator of the degree of behavioral deviation from a default policy" – If this statement were true, shouldn't the connectivity between FPl and dACC correlate with the absolute value of the difference between private and public confidence, reflecting the same rationale for using the quadratic context term as parametric regressor?

This is a very useful suggestion, and we have now revised our connectivity analysis to directly address this question. More specifically, we now include separate terms for the model-based estimate of private confidence and the empirically observed confidence reports as well as their interaction in our analysis. The advantage of this revised design matrix over using the absolute difference is that the three terms together allow us to capture signed differences in the coding of understatements (private > reported) and overstatements of confidence (private < reported). In support of our original interpretation, this analysis shows that FPl-dACC connectivity is (1) higher for larger shifts in the mapping from private to reported confidence and (2) highest when a subject understated rather than overstated their confidence. We now present this analysis in the main text (subsection “Encoding of trial-by-trial confidence in prefrontal cortex”) and have revised the figure (now Figure 5) accordingly.

1.6) There were no explanations for why there might be two significant time periods of PPI effects in Figure 4C. There are to two independent peaks, one right after the context revelation and the other, a more pronounced peak on average, occurring from 6 sec and 8 sec. Please explain these patterns, which are also seen even for the FPI-pgACC panel.

As described in response to essential revision 1.5, we have revised our connectivity analysis (now shown in Figure 5) in a way that allows us to better understand the underlying drivers of this pattern. First, around the onset of the context screen, there is a transient increase in FPl-dACC connectivity driven by the model-based estimate of private confidence (pink line). Second, 5-7s after the onset of the context screen, FPl-dACC connectivity varies as a function of not only the model-based estimate of private confidence but also the empirically observed confidence reports (cyan line) as well as their interaction (green line). In other words, the “double bump” observed in the previous Figure 4C reflected two distinct process – an early process relating to the perceptual decision and a later process relating to the social context. We now comment on both of these aspects of FPl-dACC connectivity in the main text (subsection “Encoding of trial-by-trial confidence in prefrontal cortex”).

2) Reviewers wondered if the contrast between internal and explicit confidence is contaminated by the choice of time-locking event. That is, are the results really specific to private-to-public mapping, or do they rather reflect the sequential nature of this task, and comparable findings would be observed in a number of processes involving two different states? This question has implications for the novelty of the current study. That is, if the results are not specific for private-public mapping, as opposed to a serial stage, implicit-explicit, etc. type of processes, then the novelty of the neural data would decrease profoundly (e.g., the idea of internal processing to be more medially and external processing to be more laterally localized in the PFC has been around for quite some time).

We are glad for the opportunity to expand on this issue. As we explain in response to essential revision 1.1, the choice of our time window of interest is motivated directly by the design: this is the first timepoint within a trial that context information is available. We also think there are several reasons for why the results cannot be accommodated by more generic “serial-stage” or “implicit-explicit” accounts. In a new paragraph in the Discussion (third paragraph), we provide several lines of evidence in support of a private-public mapping interpretation. In short, we note that the quadratic context term – encoded by FPl in our factorial analysis of neural activity – compares conditions that are matched in general task characteristics and only differ in the contextual requirements on a private-public mapping. Specifically, both the “inlying” contexts (medium-low and medium-high confidence) and the “outlying” contexts (low and high confidence) involve sequential preparation of a private state and a public action – the difference between these two types of context is that the latter requires a larger shift in the mapping from private to public confidence. Furthermore, we note that FPl activity was higher on the trials where the context was signalled than on the trials in which the context was hidden from subjects (now shown in Table 1). Again, these two types of trial are matched in general task characteristics and only differ in the availability of a context-dependent private-public mapping. More broadly, we believe that any task that involves explicit reports – regardless of whether the task itself involves social interaction – must to an extent invoke private-public mappings – and our results thus provide a potential explanation of why lateral PFC is often activated by such tasks.

3) Confidence model.3.1) It would be important to add a description of the model leading to confidence estimates irrespective of context, as it seems rather mysterious in the present form. Regarding the analysis strategy, please compare this modeling approach with a simpler strategy, e.g., applying mixed-effects ordinal regression on confidence with perceptual evidence and context as fixed effects.

We thank the reviewers for prompting more clarity here. As described in response to essential revision 1.2, we have substantially revised the section on neural encoding (subsection “Encoding of trial-by-trial confidence in prefrontal cortex”) – including the description of the confidence model in the main text and its associated illustration in Figure 4A. Our approach in fact very similar to the ordinal regression approach suggested by the reviewers: we (1) fit an ordinal regression model to data from the behavioural session to estimate the fixed effects of motion coherence, choice reaction time and each social context and then (2) apply the fitted model to behavioural data from the fMRI session while setting the estimated fixed effects of social context to zero. This procedure is performed separately for each subject in order to take into account individual variation in how people generally report their confidence (e.g., some people report higher confidence regardless of the task at hand; Ais et al., Cognition, 2016) – an aspect that is modelled by the “thresholds” inherent to an ordinal regression model.

3.2) In order to make sure that the confidence model and its out-of-sample prediction of private confidence estimates are reliable, it would be necessary to validate the accuracy of the model within the data it fitted to. Please report cross-validation accuracy within the data from the behavioral session.

We have included a supplementary figure (Figure 4—figure supplement 2) that shows cross-validation accuracy within the behavioural session for each subject. In short, on each iteration of a leave-one-trial-out procedure, we fit the model to all trials but one and then compute the negative log likelihood of the report observed on the left-out trial under the confidence model (where the probability distribution over reports depends on the fitted model) and a null model (where the probability distribution over reports is uniform). We then sum the cross-validated negative log-likelihoods across all trials and compute the difference between the confidence model and the null model – with a positive value indicating higher cross-validation accuracy under the confidence model than the null model. In support of our model-based approach, the confidence model outperformed the null model in all subjects.

3.3) What is the correlation between motion coherence and the model-derived estimates of private confidence? It was unclear why dACC did not encode the linear term for coherence as in Figure 3C but did track the model-derived estimates of private confidence, although both are the indirect measures of private confidence. Please explain what each of these measures differentially represent with regards to the subject's private state. Also, it was not clear why dACC would only encode one of private confidence measures.

As described in response to essential revision 1.2, we now explain the subtle but important differences between our factorial analysis of neural activity and the model-based analysis of neural encoding. More specifically, our model-based estimate of private confidence takes into account not only the impact of motion coherence on subjective experience but also the time taken to make a decision – a factor which has been shown to affect private confidence over and above the perceptual evidence itself. In addition to revising the main text, we have also added a supplementary figure that illustrates how the model-based estimate of private confidence varies with motion coherence and choice reaction time – for example, private confidence is comparable for fast decisions in response to low-coherence stimuli and slow decisions in response to high-coherence stimuli (Figure 4—figure supplement 1).

4) Reviewers were concerned that because the current study addresses a novel question with a novel approach and there was no pre-registration, that there was quite a bit of analytical freedom, for instance in the selection of ROIs, which might have biased results. To help mitigate these concerns, it would be important to (a) clearly state the a priori rational that went into the selection of ROIs, (b) fully report the results of the analyses regarding different time points (essential revision 1.1), and (c) show that the results of the ROI analyses converge with the whole-brain results. In this regard, please move the whole-brain univariate results to the main text and discuss it more thoroughly. For the RSA analysis, please conduct a searchlight to provide whole-brain results that can be compared with the ROI findings as well.

We appreciate in hindsight that formal preregistration would have been useful – especially given the specificity of our hypotheses. To provide more context for our study, and address these concerns, we have now revised the paper as follows:

a) ROI selection

We now explain the a priori reasoning behind ROI selection in a separate paragraph in the first section of the Results reporting fMRI analyses. We highlight that the three ROIs have been identified as putative neural substrates for decision confidence across a variety of studies but that their role in the generation of context-dependent confidence report is unclear. We directly hypothesised in the Discussion section of a previous paper (Bang and Fleming, PNAS, 2018) that the lateral frontopolar cortex may support a mapping from private to public confidence – a hypothesis that we suggested could be tested by manipulating the contextual requirements on a private-public mapping. Our study was also reviewed internally in January 2018 at the Wellcome Centre for Human Neuroimaging in the Centre’s “project presentation” sessions. The lateral frontopolar cortex was explicitly identified as a key ROI for mediating private-public mappings in this presentation.

b) Time window of interest

As described in response to essential revision 1.1, we now explain the reasoning behind the selection of our time window of interest.

c) Univariate whole-brain analysis

We now report the univariate whole-brain analysis of the BOLD response to the context screen in the main text (subsection “Encoding of motion coherence and social context in prefrontal cortex”).

d) Multivariate whole-brain analysis

We have now conducted a multivariate whole-brain analysis (Appendix 3 – statistical map uploaded to our NeuroVault repository). The analysis (now detailed in the Materials and methods subsection “Multivariate analysis”) is based on a first-level model using the same design matrix as our multivariate ROI analyses – modelling the whole-brain response to the context screen separately for each condition of our factorial design – but using smoothed data as we no longer extract pattern information from anatomically defined ROIs. In brief, we computed the EDI metric for the full task space (coherence x context) for a spherical cluster of voxels centred at each voxel within a subject. Consistent with previous RSA studies (e.g., Nili et al., PLOS Computational Biology, 2014), the diameter of the sphere was 15mm (around 100 voxels). The sphere was adapted in shape if it was near the edges of a whole-brain group-level mask – a mask defined as the intersection of the whole-brain masks obtained for each subject during estimation of the first-level model. Finally, whole-brain statistical testing was performed by applying one-sample t-tests against 0 to the first-level EDI maps while correcting for multiple comparisons within the SPM framework. The searchlight analysis was implemented using the RSA toolbox (Nili et al., PLOS Computational Biology, 2014) for SPM.

The searchlight analysis identified significant clusters in visual areas – a result which may be explained by the task conditions differing in visual appearance (i.e. motion coherence + colour of the avatar that indicates the current partner). We highlight that the searchlight analysis is not directly comparable to the ROI analysis but, rather, complements it. For example, the FPl ROI analysis is more sensitive to subtle pattern information as the FPl ROI contains around 10 times more voxels than the searchlight sphere. Further, our FPl ROI is bilateral and thus contains a substantial proportion of non-contiguous voxels. By contrast, the searchlight sphere is restricted to contiguous voxels within a relatively restricted region and can therefore only identify localised pattern information. In addition, our FPl ROI is anatomically defined and thus has a shape which cannot be captured by a searchlight sphere. We now explain these differences in the Materials and methods.

5) The manuscript does not contain any statistical results in the main text to support the claims. Some statistical thresholds are reported in the figure legends, but some figures (for instance Figures 4 and 5) do not contain any statistical information. Please include all statistical tests, t, f, and exact p-values in the main text.

We now provide information about statistical testing in all figure legends. We have also added tables providing statistical information for both the univariate (Table 1) and multivariate (Table 2) ROI analyses.